# Predictability and parallelism in the contemporary evolution of hybrid genomes

**Quinn K. Langdon**[1,2]*, **Daniel L. Powell**[1,2], **Bernard Kim**[1], **Shreya M. Banerjee**[1,2], **Cheyenne Payne**[1,2], **Tristram O. Dodge**[1,2], **Ben Moran**[1,2], **Paola Fascinetto-Zago**[2,3], **Molly Schumer**[1,2,4]*

**1** Department of Biology, Stanford University, Stanford, California, United States of America, **2** Centro de Investigaciones Científicas de las Huastecas "Aguazarca", A.C., Calnali, Mexico, **3** Department of Biology, Texas A&M University, College Station, Texas, United States of America, **4** Hanna H. Gray Fellow, Howard Hughes Medical Institutes, Chevy Chase, Maryland, United States of America

* qlangdon@stanford.edu (QKL); schumer@stanford.edu (MS)

## Abstract

Hybridization between species is widespread across the tree of life. As a result, many species, including our own, harbor regions of their genome derived from hybridization. Despite the recognition that this process is widespread, we understand little about how the genome stabilizes following hybridization, and whether the mechanisms driving this stabilization tend to be shared across species. Here, we dissect the drivers of variation in local ancestry across the genome in replicated hybridization events between two species pairs of swordtail fish: *Xiphophorus birchmanni × X. cortezi* and *X. birchmanni × X. malinche*. We find unexpectedly high levels of repeatability in local ancestry across the two types of hybrid populations. This repeatability is attributable in part to the fact that the recombination landscape and locations of functionally important elements play a major role in driving variation in local ancestry in both types of hybrid populations. Beyond these broad scale patterns, we identify dozens of regions of the genome where minor parent ancestry is unusually low or high across species pairs. Analysis of these regions points to shared sites under selection across species pairs, and in some cases, shared mechanisms of selection. We show that one such region is a previously unknown hybrid incompatibility that is shared across *X. birchmanni × X. cortezi* and *X. birchmanni × X. malinche* hybrid populations.

## Author summary

We now know that hybridization, or the production of offspring between individuals of different species, happens frequently across many plant, animal, and fungi groups. As a result, the genomes of many contemporary species contain material derived from these hybridization events. At the same time, hybridization can have negative consequences on an organism's ability to survive and reproduce. One major question is whether there are shared principles that determine which parts of the genome can move between species and which cannot. Here, we compare the genomes of several independently formed hybrid populations between the fish species pairs of *Xiphophorus birchmanni* and *X.*

**Data Availability Statement:** The authors confirm that all data underlying the findings are fully available without restriction. Scripts used for this project can be found in the Schumer lab github (https://github.com/Schumerlab). Scripts

specifically for these analyses can be found here: https://github.com/Schumerlab/Xbir_xcor_hybrid_genomes. FASTQ data for the populations used here are available through NCBI Sequence Read Archive. Data for the Acuapa population: PRJNA692059. Data for the Tlatemaco population: SRP130891. Data for Santa Cruz, Huextetitla, and Aguazarca here: PRJNA745218. New data for the analysis of X. birchmanni x X. malinche F2s is here: PRJNA745218 and previously published data is here: PRJNA692059. Large-scale datasets are available on dryad (doi:10.5061/dryad.4mw6m90bp).

**Funding:** This work was supported by a Knight-Hennessy Scholars fellowship and NSF GRFP (2019273798) to BM, NRSA F32 (GM135998) to BK, a CEHG fellowship and NSF PRFB (2010950) to QKL, and a Hanna H. Gray fellowship and NIH 1R35GM133774 grant to MS. The funders had no role in study design, data collection and analysis, decision to publish, or preparation of the manuscript.

**Competing interests:** The authors have declared that no competing interests exist.

*cortezi* as well as *X. birchmanni* and *X. malinche* to begin to address this question. We find that across hybrid populations, regions of the genome with especially low recombination rates or especially high gene density are more likely to be derived from the parent species that contributed the majority of the genome to hybrids. Moreover, we identify regions of the genome that negatively interact in both types of hybrid population. Together our results demonstrate that genomic outcomes of hybridization between distinct species pairs are in part predictable.

## Introduction

In the past decade the number of species with sequenced genomes has risen exponentially [1–3]. This has allowed for a greater exploration of the evolutionary histories of the diversity of life on the planet. One consequence of this work has been a new understanding of the ubiquity of genetic exchange between species [4–6]. Studies have demonstrated the impact of both ancient and recent hybridization on the genomes of extant species from fruit flies to humans [7–14]. While it is now clear that hybridization is commonplace, what is less understood is how the genome changes after admixture and what genetic and evolutionary factors generally shape this process.

An important piece of predicting how the genome will respond after hybridization is understanding the sources of selection that act on hybrids. A large number of studies have shown that hybrids between different species tend to experience fitness consequences; they are often less capable of surviving and reproducing than the species that formed them [15–20]. Several types of selection on hybrids can drive this pattern. Decades of empirical and theoretical work have indicated that Dobzhansky-Muller hybrid incompatibilities (DMIs), or incompatible interactions between mutations that are derived in each of the parental species, are a common cause of inviability and infertility in hybrids [21–24]. Hybrids can also suffer from the effects of intermediate or transgressive traits that place them far from the phenotypic optimum for either parental species [25–28]. In both of these scenarios, selection is generally expected to act against alleles derived from the "minor" parent species, or the species that contributed less to the genome of the hybrids. This is because removing incompatible alleles derived from the minor parent species (or variants that move a hybrid further away from the major parent's phenotypic optimum) is the fastest route to restoring fitness [6]. In addition, recent work has indicated that another mechanism of selection against hybrids can stem from differences in historical effective populations size between the parental species. Such differences can result in the disproportionate accumulation of deleterious mutations along the parental lineage with the smaller historical effective population size and drive the removal of haplotypes from this parental species after hybridization [19, 29]. However, this mechanism is most likely to impact hybrids that form between species with dramatically different historical effective population sizes [19, 29].

Although it is well-established that hybrids between species frequently experience selection, it is less clear how predictable the outcomes of selection are at a genome-wide scale, both within and between species pairs. While it has been appreciated for decades that certain regions of the genome, such as sex chromosomes, tend to retain less minor parent ancestry [14, 30–33], recent studies have highlighted other patterns that appear to be consistent across several independent hybridization events. These include broad selection against minor parent ancestry across the genome (presumably due to the mechanisms outlined above) and stronger depletion of minor parent ancestry in regions of the genome with a high density of conserved

or coding elements [8, 12, 34, 35]. Another key variable highlighted by empirical and theoretical studies is the importance of the interplay between recombination rate and selection, with several groups finding that selection on hybrids generates a positive correlation between local recombination rate and minor parent ancestry [8, 34–40]. This correlation is thought to be driven by how the local recombination rate impacts the decoupling of neutral or adaptive alleles from alleles that are deleterious in hybrids. Neutral or adaptive minor parent alleles that occur in regions of the genome with high local recombination rates are less likely to remain linked to deleterious neighbors, resulting in the retention of more minor parent haplotypes in such regions after selection.

These emerging findings highlight how certain features of genome organization can interact with selection after hybridization in a predictable way across diverse species [6]. This suggests that we can expect some broad scale repeatability in the outcomes of hybridization across species and the types of regions of the genome that are more or less permeable to introgression. In hybridization events between pairs of closely related species we may expect repeatability in the outcomes of genome evolution after hybridization to be even higher. This is in part because close relatives are expected to share genome composition and conserved local recombination maps (in some species; [41, 42]). However, in close relatives, repeatability in the outcomes of hybridization could also be driven by shared mechanisms of selection acting on individual sites. For example, close relatives may share DMIs due to shared phylogenetic history [43] or similarities in underlying genetic networks [44] and among some species groups, certain alleles may be globally adaptive, leading to shared ancestry driven by adaptive introgression. Thus, leveraging comparisons of independent natural hybridization events between closely related species pairs is one promising approach for evaluating the factors that drive repeatable evolution after hybridization [12].

Recently formed hybrid populations are an especially powerful tool for addressing these questions because ancestry variation along the genome can reflect the action of strong selection on hybrids. Here, we leverage swordtail fish species (genus *Xiphophorus*) to investigate genome evolution after several recent, independent hybridization events. Natural replicate hybrid populations formed between *X. birchmanni* and *X. malinche* are an emerging model system for studying genome evolution after hybridization. Research on this sister species pair has shown that variation in local ancestry along the genome is best explained by the presence of prevalent hybrid incompatibilities (as opposed to historical effective populations size differences in the parental species or ecological drivers of ancestry variation; [34]), and has documented strong DMIs between them [23]. Here, we expand on our recent description of replicate hybrid populations of the species pair *X. birchmanni* × *X. cortezi* [45] to explore repeatability in local ancestry between these two types of hybrid populations.

*X. birchmanni* × *X. cortezi* are more deeply diverged than *X. birchmanni* × *X. malinche* (~450k generations diverged and ~250k generations diverged respectively) and unlike *X. birchmanni* and *X. malinche*, they do not differ in their time-averaged historical effective population sizes (S1 Fig) [45]. Hybrid populations between both species pairs formed recently (~100–150 generations ago [20, 45]), likely due to human disturbance [46]. Comparisons across these two types of hybridization events provide one of the first windows into the drivers of evolution after hybridization in related species, allowing us to address questions about predictability and parallelism in selection on hybrids at both a local and broad-scale along the genome.

## Results

### Background

Throughout the manuscript we explore repeatability in genome evolution across two types of hybrid populations, those formed between the *X. birchmanni* × *X. cortezi* species pair and

those formed between the *X. birchmanni* × *X. malinche* species pair. These three species are closely related, with divergence between *X. birchmanni* × *X. cortezi* slightly higher than divergence between *X. birchmanni* × *X. malinche* (Fig 1; sequence divergence 0.6% and 0.4% per basepair respectively). Nucleotide diversity and inferred long-term effective population sizes are similar between *X. birchmanni* and *X. cortezi* (π ~ 0.1% per basepair; [45]) and lower in *X. malinche* (π ~ 0.03% per basepair; [34]) (S1 Fig). *X. birchmanni* is distributed between *X. malinche* and *X. cortezi*, with hybridization occurring in areas of the rivers where the species are

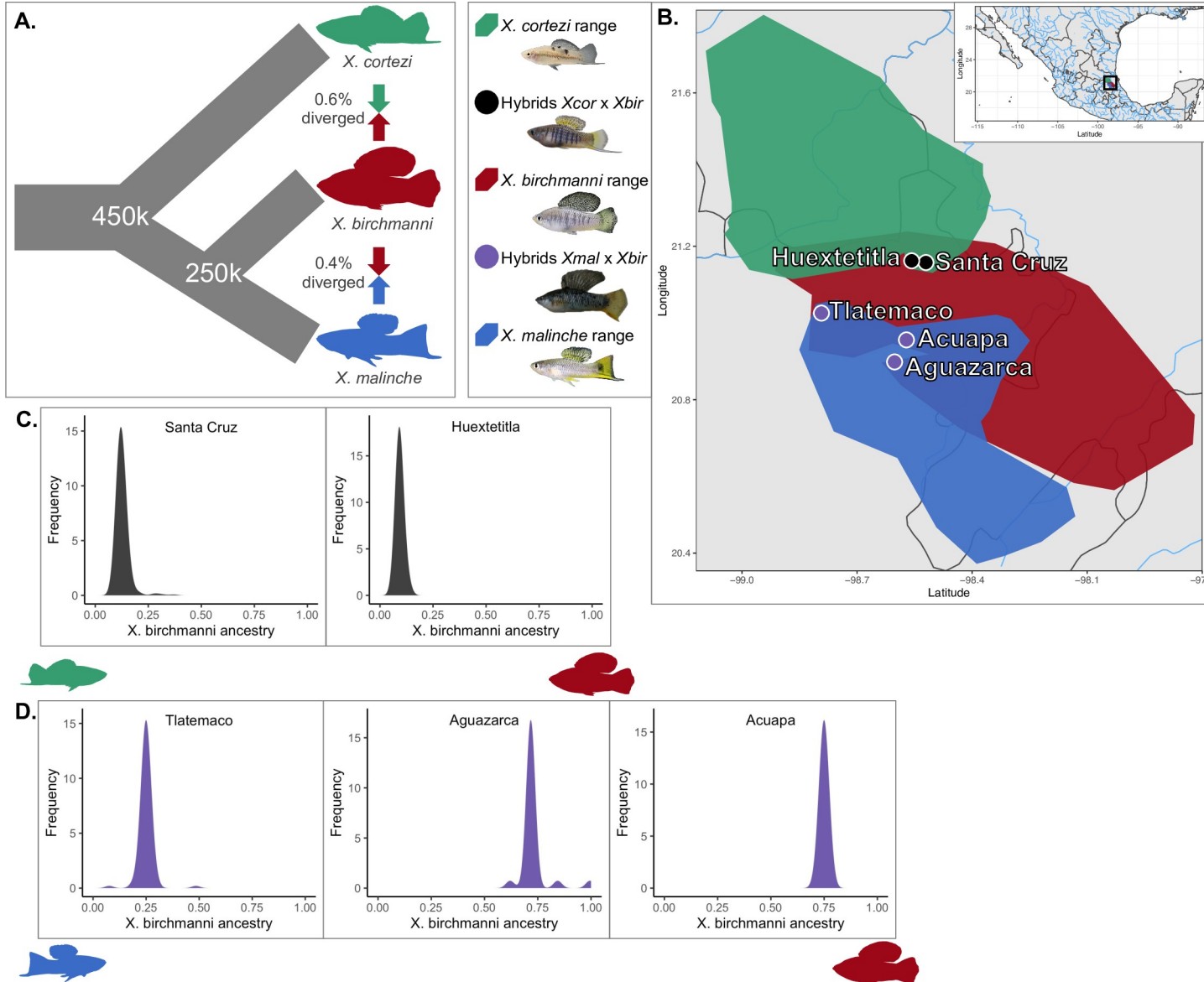

**Fig 1. Description of focal species, study sites, and hybrid population ancestry structure. A.** Phylogenetic relationships and contemporary hybridization events between focal species *X. birchmanni*, *X. malinche*, and *X. cortezi*. **B.** Distributions of *X. birchmanni* (*Xbir*), *X. malinche* (*Xmal*), and *X. cortezi* (*Xcor*), and their hybrids in Hidalgo and San Luis Potosí, Mexico. Blue shading shows the known range of *X. malinche*, red shading shows the known range of *X. birchmanni*, and green shading shows the known range of *X. cortezi*. **C.** Genome-wide admixture proportions of *X. birchmanni* × *X. cortezi* hybrids found in the Santa Cruz and Huextetitla replicate hybrid populations. **D.** Genome-wide admixture proportions of *X. birchmanni* × *X. malinche* hybrids found in the replicate populations at Tlatemaco (N = 95), Aguazarca (N = 51), and Acuapa (N = 97). Map base Natural Earth (ne_10m_admin_1_states_provinces, ne_10m_lakes, ne_10m_lakes_north_america, ne_10m_rivers_lake_centerlines, and ne_10m_rivers_north_america).

sympatric (Fig 1); *X. malinche* and *X. cortezi* are allopatric throughout their ranges. We have previously shown that replicate hybrid populations in both species pairs formed recently, within the last ~100–150 generations [20, 34, 45, 47], likely due to human-mediated habitat disturbance [46].

For clarity, we refer to the hybridization events between each pair of species as the two "types" of hybridization events. In addition, we analyzed data from two *X. birchmanni* × *X. cortezi* (Huextetitla and Santa Cruz) and three *X. birchmanni* × *X. malinche* (Acuapa, Aguazarca, and Tlatemaco) hybrid populations, each of which occur in different rivers or tributaries (Fig 1). When discussing comparisons within a hybridization type (e.g. *X. birchmanni* × *X. cortezi* or *X. birchmanni* × *X. malinche*) we refer to these geographically distinct samples as "replicate" hybrid populations. Past work in the *X. birchmanni* × *X. malinche* system has shown that the replicate hybrid populations formed independently [48]. Additionally, they also differ in their genome-wide ancestry: *X. birchmanni* is the minor parent in Tlatemaco, while *X. malinche* is the minor parent in Acuapa and Aguazarca (Fig 1).

Despite the fact that similar factors (i.e. population disturbance) have likely driven recent hybridization across the two species pairs [45, 46], each replicate population occurs in a distinct ecological environment, and varies in the degree of assortative mating observed [34, 45, 47, 48]. For a more detailed description of what is known about the ecological environments and mating dynamics of each population from past work see Text A in S1 File.

The Santa Cruz and Huextetitla hybrid populations occur along geographically separated tributaries of the same river (Fig 1) and have similar genome-wide admixture proportions and ancestry structure [45]. Hybrids in both populations derive 85–90% of their genomes from the *X. cortezi* parent species (Fig 1). Using a hidden Markov model (HMM) based approach, we collected low-coverage whole genome sequence data and inferred local ancestry across the genome in 254 hybrids from the two populations (see Methods). While we initially planned to use the two sites to explore repeatability in genome evolution within the replicate populations, we found evidence that they were not completely independent hybridization events (S2 and S3 Figs and Text B in S1 File). Thus, we focus our discussion on the Santa Cruz hybrid population but present analyses for the Huextetitla hybrid population (for which results are qualitatively similar) in parts of the main text and supplement.

## Inference of demographic histories of hybrid populations

Initial work investigating the demographic history of the *X. birchmanni* × *X. cortezi* hybrid populations at Santa Cruz and Huextetitla [45] suggested that they formed recently and experience low levels of migration from the parental species [45]. In this paper we expanded our sampling from previous work [45] and explored the demographic history of *X. birchmanni* × *X. cortezi* and *X. birchmanni* × *X. malinche* hybrid populations in more detail. We used simulations and an approximate Bayesian computation approach (ABC, Text C in S1 File) to infer the demographic history of two populations: the Santa Cruz and Acuapa hybrid populations. Drawing from uniform (or log-uniform) prior distributions for parameters of interest (initial admixture proportions, time since initial admixture, population size, and migration rates), we conducted simulations in SLiM and performed rejection sampling based on summary statistics derived from the observed data (Text C in S1 File). These simulations yielded well-resolved posterior distributions for most parameters in both hybrid populations (S4 Fig). We discuss these results in more detail in Text C in S1 File. Briefly, our findings are consistent with our previous work [45]; both hybrid zones are young and experience limited migration from the parental species, but the *X. birchmanni* × *X. cortezi* hybrid population formed less recently (~120 generations ago) than the *X. birchmanni* × *X. malinche* (~80 generations ago) hybrid

population (S4 Fig). Together, our ABC results allow us to model plausible demographic histories for each type of hybrid population to explore expected patterns of genome evolution.

### Predictable genomic features correlate with minor parent ancestry in *X. birchmanni × X. cortezi* hybrid populations

Although admixture began recently between *X. birchmanni* and *X. cortezi*, we observe substantial heterogeneity in local ancestry along the genome, with regions fixed for both major and minor parent ancestry (e.g. S3 Fig). While severe bottlenecks could also generate fixation in recently formed hybrid populations [45], this variance in local ancestry could also be indicative of selection on hybrids. Indeed, simulations of variation in local ancestry using the demographic parameters we infer with ABC (admixture time, admixture proportion, and hybrid population size) suggest that the patterns of fixation in major and minor parent ancestry we observe are unexpected under the most likely scenario of neutral admixture (S5 Fig and Text C in S1 File).

We next evaluated correlations between minor parent ancestry and genomic features known to interact with selection in other hybrid systems. Studies in a number of systems have found that selection on hybrids can drive a positive correlation between recombination rate and minor parent ancestry [8, 34, 35, 39]. Using averages of minor parent ancestry and recombination rate summarized in non-overlapping windows we found strong evidence that recombination rate is positively correlated with minor parent ancestry at both a local (Fig 2; Santa Cruz– 100 kb windows: $\rho = 0.44$, $p < 10^{-300}$; Huextetitla– 100 kb windows: $\rho = 0.42$, $p < 10^{-275}$) and chromosome-wide scale (Text D in S1 File and S6 and S7 Figs) in replicate hybrid populations of *X. birchmanni × X. cortezi*. This pattern is robust across spatial scales (S1 Table) and when controlling for a number of possible technical variables; e.g. thinning windows based on linkage disequilibrium and use of recombination rate estimates from observed crossovers in $F_2$s (see S2 and S3 Tables).

We also investigated how ancestry in *X. birchmanni × X. cortezi* hybrid populations covaries with coding and conserved basepairs across the genome. Based on our previous findings in *X. birchmanni × X. malinche* hybrid populations [34], we predicted that regions with more coding and conserved basepairs would be depleted in minor parent ancestry. We summarized average minor parent ancestry and number of coding and conserved basepairs in non-overlapping genetic windows, to account for variation in recombination rate, and found depletion of minor parent ancestry in regions especially dense in coding and conserved basepairs (Fig 2 and S4 Table; Santa Cruz– 0.1 cM windows coding basepairs: $\rho = -0.15$, $p < 10^{-73}$, conserved basepairs: $\rho = -0.22$, $p < 10^{-161}$; Huextetitla– 0.1 cM windows: coding basepairs: $\rho = -0.09$, $p < 10^{-27}$, conserved basepairs: $\rho = -0.16$, $p < 10^{-79}$). This pattern is robust across a range of window sizes (S4 Table), and when controlling for a number of potential technical artifacts (S5–S9 Tables). Interestingly, we did not detect an impact of nonsynonymous substitutions between species on minor parent ancestry after controlling for local substitution rates (Text E in S1 File and S8 Fig).

### Sources of selection driving variation in minor parent ancestry

Above we confirmed via simulations that the observed variation in local ancestry is not expected by chance or as a consequence of the demographic history of hybrid populations (Text C in S1 File), motivating us to ask about the role different sources of selection might play in driving the broad-scale ancestry patterns we observed in *X. birchmanni × X. cortezi* type and *X. birchmanni × X. malinche* type hybrid populations. To do so, we used simulations,

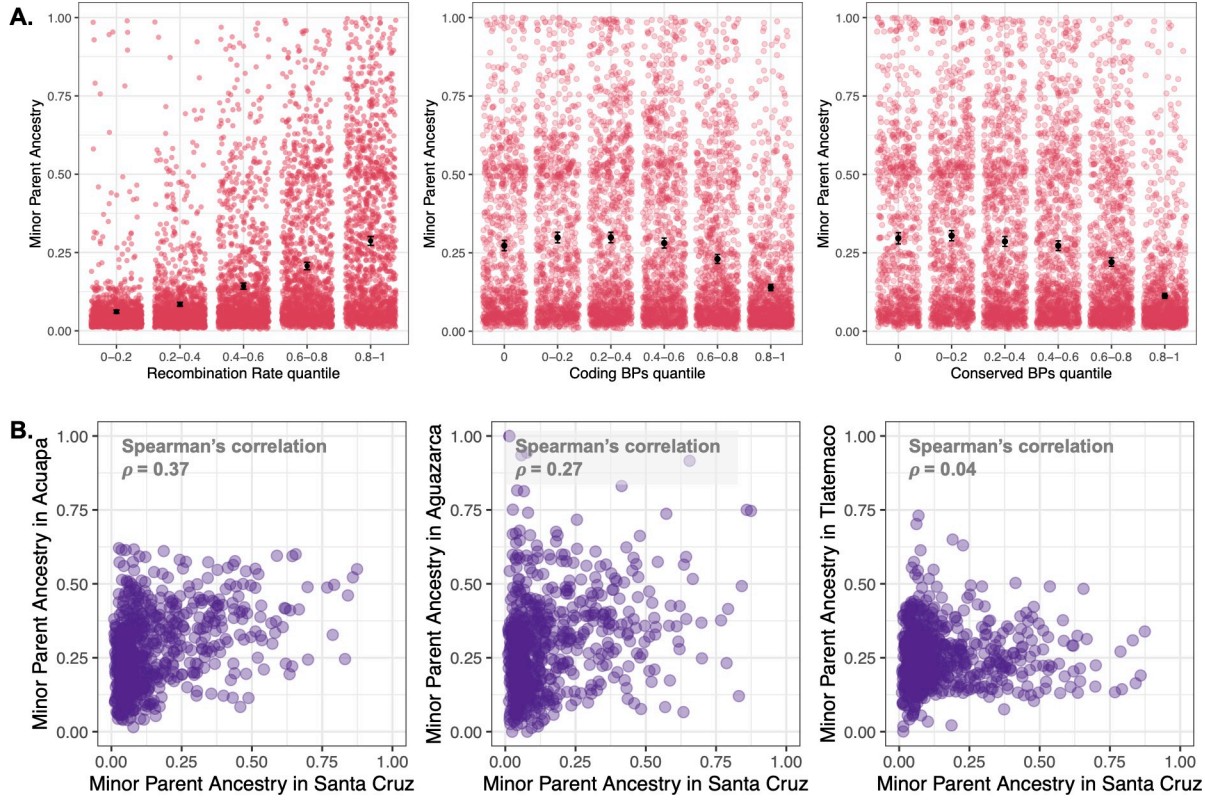

**Fig 2. Relationship between minor parent ancestry, recombination rate and functional basepairs in *X. birchmanni* × *X. cortezi* hybrid populations, as well as cross-population type comparisons. A.** Minor parent ancestry (*X. birchmanni* ancestry) in the Santa Cruz hybrid population is higher in regions of the genome with higher recombination rates. Semi-transparent points show results from individual 100 kb non-overlapping windows, black points and whiskers show the mean ± two standard errors of the mean. After accounting for recombination rate by averaging ancestry in 0.25 cM non-overlapping windows, we also observe a strong negative correlation between minor parent ancestry and the number of linked coding (middle) and conserved basepairs (right). We find the same results when analyzing data from the Huextetitla hybrid population and across a range of window sizes (S1 and S4 Tables). Note that results for **A** are shown in quintiles for visualization but statistical tests reported in the text were performed on unbinned data. **B.** Correlations in local ancestry, plotted here in 1 Mb non-overlapping windows. Cross-population ancestry correlations between Santa Cruz (or Huextetila–S10 Table) and *X. birchmanni* × *X. malinche* hybrid populations are significant in all comparisons except for the majority *X. malinche* population Tlatemaco (far right; see main text & Text F in S1 File). Some of the ancestry covariance between *X. birchmanni* × *X. cortezi* and *X. birchmanni* × *X. malinche* hybrid populations can be explained by shared features of genome architecture such as the locations of coding and conserved basepairs and the local recombination rate, but substantial covariance remains after accounting for these features (S12 Table).

grounded in the inferred demographic history of the hybrid populations (Texts C and F in S1 File).

In our empirical data, minor parent ancestry is consistently depleted in regions of the genome with low recombination rate (and higher coding/conserved density), regardless of *which* species is the minor parent in a given population. Our simulations indicate that these positive correlations between minor parent ancestry and recombination rate across replicate populations and species pairs are best explained by a model where selection acts against minor parent ancestry at many sites across the genome. In simulations we use a model of selection against hybrid incompatibilities, which results in selection against minor parent ancestry at incompatible sites and has been well-documented in swordtails [20, 23, 34, 49]. We note, how-ever, that a subset of other models of selection on hybrids could generate similar results (see discussion in Text F in S1 File; reviewed in [6]). However, the patterns we observe are incon-sistent with models of selection against hybridization load or global selection against one of the parental species (S9 Fig and Text F in S1 File). This latter finding indicates that selection

on the *X. birchmanni* genome, which is a shared parent species in both hybrid population types, does not underlie the patterns that we document.

## Repeatability extends across hybrid population types

Given evidence for selection against minor parent ancestry in hybrid populations of both types, the key question we wish to understand is how repeatable the outcomes of admixture are across different species pairs, and what mechanisms contribute to this repeatability. Past work has suggested that we expect repeatability in local ancestry in replicate hybrid populations formed between the same species pairs [34, 50]. This repeatability often exceeds expectations from shared genetic architecture (e.g. recombination rate and functional basepair density), indicating that selection on the same sites in replicate hybrid populations drives concordance in local ancestry [34, 50].

We wanted to evaluate whether there was evidence for a similar phenomenon across the two distinct types of hybridization events studied here. The *X. birchmanni* × *X. cortezi* hybridization events are independent from each of the three *X. birchmanni* × *X. malinche* hybrid populations (Fig 1). In addition, the three replicate *X. birchmanni* × *X. malinche* hybrid populations we analyze (Acuapa, Aguazarca, and Tlatemaco) occur in geographically isolated river systems (Fig 1) and formed independently of each other [48].

We observe repeatability in local patterns of minor parent ancestry across the genome between most pairs of hybrid populations formed between these distinct species pairs (in 100 kb non-overlapping windows $\rho_{\text{Santa Cruz—Acuapa}} = 0.25$, $p<10^{-93}$; $\rho_{\text{Santa Cruz—Aguazarca}} = 0.17$, $p<10^{-42}$; Fig 2 and S10 Table), an intriguing observation given the demographic independence of these hybrid populations. These results were robust to a number of factors, including accounting for the locations of structural differences between the species (see S11 Table and Texts D and F in S1 File). One exception to this pattern is in comparisons involving the Tlatemaco population. We speculate that this is driven by the fact that hybrids in this population derive the majority of their genome from *X. malinche*, a species with low historical effective population sizes (S1 Fig). We discuss this pattern in more detail in Text F in S1 File.

Because many features of the architecture of the genome, such as the local recombination rate and the locations of coding and conserved basepairs, are largely conserved across *X. birchmanni*, *X. malinche*, and *X. cortezi* (Text D in S1 File), we wanted to evaluate whether cross-population repeatability exceeded what was expected from these features alone. Partial correlation analysis of local ancestry across pairs of populations, including these features as covariates, indicated that some of the signal of shared local ancestry across hybrid population types can be explained by these features (S12 Table). However, after accounting for this shared genomic architecture, we still observed repeatability in patterns of minor parent ancestry between most pairs of hybrid populations, including those formed between distinct species pairs (in 100 kb non-overlapping windows $\rho_{\text{Santa Cruz—Acuapa}} = 0.15$, $p<10^{-33}$; $\rho_{\text{Santa Cruz—Aguazarca}} = 0.09$, $p<10^{-15}$; S12 Table). Simulations indicate that the observed correlations in local ancestry across the two types of hybridization events are unlikely to be driven by demographic history but can readily be generated by shared sites under selection (S10 Fig and Texts C and F in S1 File).

## Shared islands and deserts of minor parent ancestry across hybrid population types

The analyses described above indicate that not all cross-population repeatability in ancestry can be explained by shared genomic architecture between these closely related species pairs (S10, S12 and S13 Tables and Text F in S1 File). We next explored how other factors, such as

shared sites where minor parent ancestry is favored or disfavored, contributed to repeatability in local ancestry. Specifically, we asked whether there were certain regions of the genome with shared patterns of minor parent ancestry across hybrid population types, and whether we could identify the mechanisms of selection driving these shared patterns.

We first focused on the Santa Cruz *X. birchmanni* × *X. cortezi* population to identify regions of the genome with unusual minor parent ancestry. We identified minor parent "deserts" by scanning the genome for ancestry informative sites that fell in the lower 2.5% tail of genome-wide ancestry. We then expanded out from these sites to define regions of reduced minor parent ancestry (defined as those falling in the lower <5% tail of genome-wide ancestry; see Methods). These regions were then filtered to exclude very short ancestry tracts, those supported by few ancestry informative sites, and were merged with nearby deserts (see Methods and S11 Fig). This procedure resulted in the identification of 81 minor parent deserts with an average length of 317 kb spread across 18 of the 24 swordtail chromosomes. Of these 81 minor parent ancestry deserts, 21 (26%) overlapped with regions of low minor parent ancestry in one or more replicate *X. birchmanni* × *X. malinche* hybrid population (see Methods).

Using the same approach, we identified 76 regions of unusually high minor parent ancestry (minor parent ancestry "islands"), in the Santa Cruz *X. birchmanni* × *X. cortezi* population, where minor parent ancestry fell in the upper 2.5% tail of the genome wide ancestry distribution. Approximately 25% of these minor parent islands were also identified in one or more *X. birchmanni* × *X. malinche* hybrid population.

We next explored whether the amount of sharing of both deserts and islands exceeded what would be expected by chance using two approaches. First, we randomly sampled 0.05 cM windows from each focal population and counted the number of windows overlapping with detected deserts and islands that had low or high minor parent ancestry. With this approach we found that sharing of both low and high minor parent ancestry regions in the real data exceeded sharing expected by chance (Fig 3A; Methods; both deserts and islands p<0.001 by permutation). Because this simple approach could introduce artifacts by disrupting autocorrelation in local ancestry along the genome, we also evaluated expected ancestry sharing by chance using an approach where we preserved the structure of local ancestry by shuffling the genome in large chunks (Text H in S1 File and S12 Fig; deserts p<0.008, islands p = 0.2, by permutation). While both approaches indicated that the number of shared minor parent deserts in the real data is unexpected by chance, support for shared islands is weaker.

We also evaluated shared ancestry deserts and islands in light of a number of technical factors that could bias ancestry calls and potentially drive artifacts in the data, such as variation in power to infer ancestry along the genome and coincidence of minor parent deserts and islands with repetitive regions (Text H in S1 File). We do not find evidence for systematic differences in power or expected error rates between shared and unshared regions (S13 and S14 Figs and Text H in S1 File). We also found that shared minor parent deserts and islands are not unusual in the number of conserved or coding basepairs they contain, suggesting that they are not driven by regions with an especially high or low density of conserved or coding basepairs (Text H in S1 File and S15 Fig). Moreover, minor parent deserts were still ancestry outliers when compared to regions of the genome predicted to have especially high constraint and minor parent islands remained ancestry outliers when compared to regions of the genome predicted to have especially low constraint (S16 Fig and Text H in S1 File). Finally, simulations indicated that the approach we use to identify shared minor parent deserts has good power even at moderate selection coefficients (e.g. *s*~0.05; Text G in S1 File). Overall, our analyses suggest that our approach to identifying regions of high and low minor parent ancestry between hybrid population types is robust to technical artifacts.

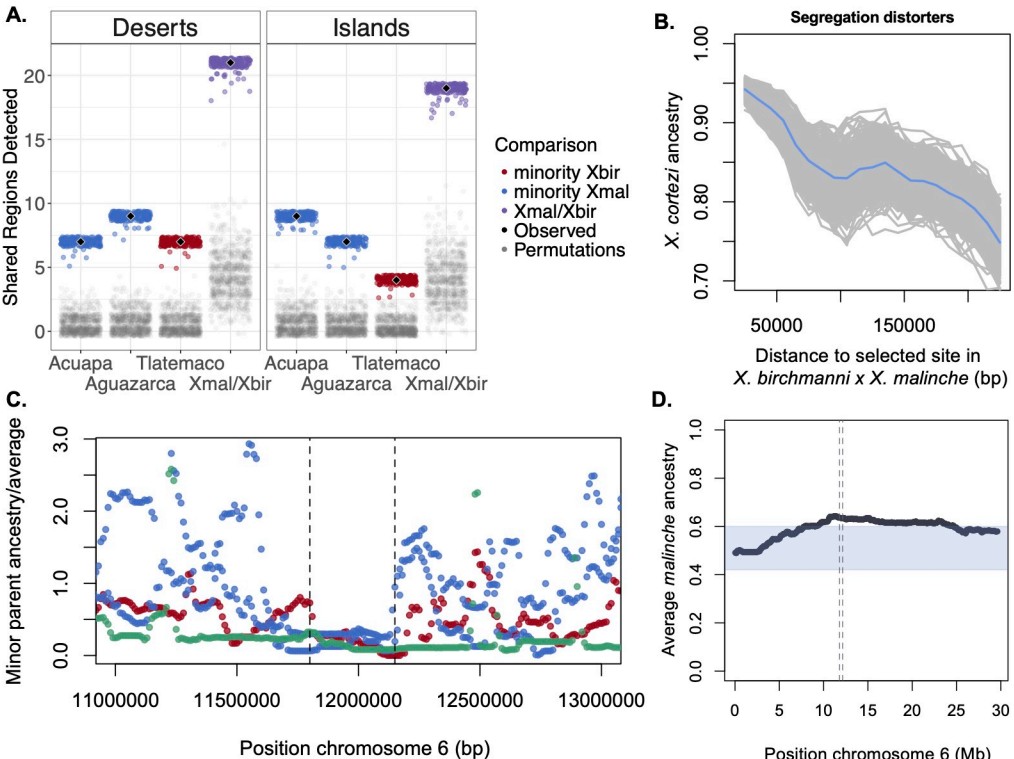

**Fig 3. A.** Shared minor parent deserts and islands are enriched compared to expectations by chance between *X. birchmanni* × *X. cortezi* and *X. birchmanni* × *X. malinche* hybrid populations. Results shown here indicate the number of shared minor parent deserts (or islands) between the Santa Cruz *X. birchmanni* × *X. cortezi* hybrid population and each *X. birchmanni* × *X. malinche* hybrid population (Acuapa, Aguazarca, and Tlatemaco). Black diamonds show the observed number of shared minor parent deserts or islands and colored points show the results when the data is jack-knife bootstrapped. In contrast, few shared minor parent deserts or islands are expected by chance (results of permutations shown by gray points). The column labeled Xmal/Xbir shows the number of shared deserts or islands between the Santa Cruz *X. birchmanni* × *X. cortezi* hybrid population and any single *X. birchmanni* × *X. malinche* population. See Text H in S1 File and S12 Fig for the results of alternative analysis approaches. **B.** Minor parent ancestry in *X. birchmanni* × *X. cortezi* hybrid populations is depleted near 6 segregation distorters identified in artificial crosses between *X. birchmanni* × *X. malinche* (see Results). Results plotted here are for the Santa Cruz hybrid population; gray lines show 500 replicates bootstrap resampling windows in the data, blue line indicates the mean across simulations. **C.** One minor parent ancestry desert on chromosome 6 was detected across all *X. birchmanni* × *X. cortezi* and *X. birchmanni* × *X. malinche* hybrid populations. Points represent minor parent ancestry calculated in 10 kb non-overlapping windows divided by the proportion of the genome derived from the minor parent species. Red–Tlatemaco (*malinche* × *birchmanni*), Blue–Acuapa and Aguazarca (*birchmanni* × *malinche*), Green–Santa Cruz (*birchmanni* × *cortezi*). **D.** The shared minor parent ancestry desert on chromosome 6 coincides with a strong segregation distorter in artificial crosses between *X. birchmanni* × *X. malinche*. Plotted here is average ancestry at ancestry informative sites across 943 $F_2$ hybrids, where expected ancestry is 50% *X. birchmanni* and 50% *X. malinche*. The blue envelop indicates the 99% quantile of variation in ancestry in this cross genome-wide. The dashed lines indicate the location of the shared ancestry desert in hybrid populations plotted in **C**.

## Features coinciding with shared minor parent ancestry deserts and islands

The unexpectedly high number of shared minor parent ancestry deserts and islands is consistent with a model where there are some shared loci under selection across species pairs. These shared ancestry patterns could be driven by a number of shared sources of selection on hybrids across the two species pairs, including selection against hybrid incompatibilities, shared signatures of adaptive introgression, among other possible mechanisms (Text F in S1 File). We evaluated the locations of minor parent deserts and islands across the two types of hybrid populations to begin to explore the mechanisms of selection driving them. Two of the 21

shared minor parent deserts overlap with known segregation distorters in *X. birchmanni* × *X. malinche* that are likely caused by hybrid incompatibilities (see next section), and three shared deserts correspond to structural differences between species on chromosome 7 and chromosome 17. Another shared desert was found in all replicate hybrid populations where *X. birchmanni* is the minor parent. This region spans the first two megabases of chromosome 21, the putative sex chromosome (S17 Fig), and is depleted in *X. birchmanni* ancestry. However, consistent with previous work in swordtails, we do not detect an enrichment of shared minor parent deserts (or islands) on chromosome 21 (Text H in S1 File), possibly because sex chromosomes are young and not strongly differentiated between males and females [51].

Of the 19 regions with especially high minor parent ancestry across hybrid population types, two correspond to an inversion that occurs on chromosome 17 and one corresponds to an inversion on chromosome 19 (S18 Fig). Intriguingly, inversions that differed between species had unusually high minor parent ancestry in all hybrid populations on average (S19 Fig), in contrast to what is most frequently observed in other systems ([52–54] but see [39, 55, 56]).

The remaining minor parent deserts and islands do not coincide with known structural differences between species [34, 45], mapped hybrid incompatibilities between *X. birchmanni* × *X. malinche* [20, 23, 49], or mapped QTL underlying trait differences between the species [23, 57]. We investigate other features associated with these regions below.

## Ancestry at sites under selection in *X. birchmanni* × *X. malinche* hybrids

We observe some overlap between known hybrid incompatibilities and minor parent ancestry deserts, suggesting that this is one mechanism that can drive cross-population repeatability in ancestry (among several plausible mechanisms; Text F in S1 File). To test the role of hybrid incompatibilities in driving cross-repeatability more formally, we took advantage of sites likely to be involved in hybrid incompatibilities in replicate *X. birchmanni* × *X. malinche* hybrid populations and asked if they had particularly low minor parent ancestry in *X. birchmanni* × *X. cortezi* hybrid populations.

Nearly a decade of work has aimed to identify hybrid incompatibilities between *X. birchmanni* × *X. malinche* using natural hybrid populations and artificial crosses [20, 23, 34, 49]. We focused on two available datasets: putative segregating DMIs identified in natural hybrid populations between *X. birchmanni* × *X. malinche* [20, 49] as well as six segregation distorters identified in $F_2$ hybrids of lab crosses between *X. birchmanni* × *X. malinche* that show ancestry patterns consistent with the action of strongly selected DMIs (e.g. $s > 0.25$; see Methods; S20 Fig). For the segregation distorter dataset, we found that sites under selection in *X. birchmanni* × *X. malinche* $F_2$ hybrids had lower minor parent ancestry on average in the replicate *X. birchmanni* × *X. cortezi* hybrid populations (Fig 3B). Moreover, we show that this result is robust to excluding a segregation distorter that is also a shared minor parent ancestry desert across all populations (S20 and S21 Figs). We do not see an effect of DMIs mapped in *X. birchmanni* × *X. malinche* natural hybrid populations that were inferred to be under weaker selection, which could indicate that they are not enriched for hybrid incompatibilities between *X. birchmanni* × *X. cortezi* or that we have low power to detect incompatibilities under weak selection [20].

## Mapping a newly identified shared hybrid incompatibility

In the previous section we showed that hybrid incompatibilities identified in one species pair are enriched for low minor parent ancestry in natural hybrid populations formed between another species pair, indicating that some hybrid incompatibilities are repeatable across the two types of hybrid populations. One striking signal we observe in our data is the coincidence of an ancestry desert on chromosome 6 shared across all populations where *X. birchmanni* is

the minor parent and a strong segregation distorter identified in $F_2$ hybrids between *X. birchmanni × X. malinche* (Fig 3C and 3D; see Methods). Further examination of this region indicated that it is depleted for minor parent ancestry in all replicate *X. birchmanni × X. malinche* populations (Fig 3C and 3D, though it did not reach the shared desert threshold in all populations).

Since minor parent ancestry in this region appears to be deleterious across all hybrid populations in both species pairs, we predicted that it was likely to be a shared hybrid incompatibility ([34]; Text F in S1 File). Epistatic hybrid incompatibilities, by definition, involve two or more loci, so we set out to identify loci interacting with the chromosome 6 region in hybrids. Because this region appears to be under selection in *X. birchmanni × X. malinche* $F_2$ hybrids, we leveraged our data from *X. birchmanni × X. malinche* $F_2$s to scan for interacting loci. Using deviations from expected two locus genotypes (see Methods), we mapped a locus that interacts with the chromosome 6 region to chromosome 13 (Fig 4A). Several genotype combinations are found in $F_2$ hybrids at lower than expected frequencies (Fig 4).

Examining the chromosome 13 region in natural hybrid populations of both *X. birchmanni × X. malinche* and *X. birchmanni × X. cortezi*, we found that it also tended to have lower minor parent ancestry across populations than expected by chance, although it was not extreme enough to have been identified as a shared desert in our initial scan. At the center of the associated chromosome 13 region, minor parent ancestry in four out of five of the hybrid populations fell into the lower 10% quantile genome-wide (S22 Fig; empirical p-value based on ancestry in 10 kb non-overlapping windows–p<0.005).

The associated regions on chromosome 6 and chromosome 13 identified in $F_2$ hybrids between *X. birchmanni × X. malinche* contain 13 and 29 genes respectively. We used the STRING database [58] to evaluate whether any pairs of genes across the two regions are known to interact, restricting interaction criteria to those supported with experimental evidence, evidence of dimerization, or evidence of co-expression. Only one pair of genes was annotated as interacting based on these criteria across the two regions on chromosomes 6 and 13, the mitochondrial complex I genes *ndufs5* and *ndufa13*. Both these genes are strongly conserved in other *Xiphophorus* species [59], however several nonsynonymous substitutions distinguish these three species and appear to be derived in *X. birchmanni* (Table 1). Intriguingly, all of the hybrid populations analyzed have fixed the mitochondrial haplotype of their major parent species (S23 Fig; see also [59]), and our cross setup means that all $F_2$ hybrids had *X. malinche* mitochondria since the reverse cross is rarely successful (see Methods). We explore the possibility that these genes form a complex incompatibility with mitochondrial interactors in a companion study [59].

Similar analyses provided some evidence for the presence of additional shared hybrid incompatibilities involving other shared minor parent deserts (see Methods; Text I in S1 File). These included evidence for another pairwise hybrid incompatibility involving a minor parent ancestry desert on chromosome 9 and an interacting locus on chromosome 4 (Fig 4C and 4D) and evidence for a complex hybrid incompatibility involving a minor parent ancestry desert identified on chromosome 5 (S24 Fig and Text I in S1 File), both detected at FPR threshold of 10%.

Evidence from simulations indicated that in contrast to good power to detect minor parent deserts in the population data driven by moderate selection ($s\sim0.05$), we have poor power to detect interactions in the $F_2$ dataset, with the exception of cases in which selection is extremely strong (e.g. $s > 0.25$; Texts G and I in S1 File). Given that we likely lack power to detect many interactions between shared minor parent deserts and other regions of the genome, we also analyzed the data using a broader approach. We asked whether there was evidence that genes in minor parent deserts (or islands) were enriched for known protein-protein interactions or

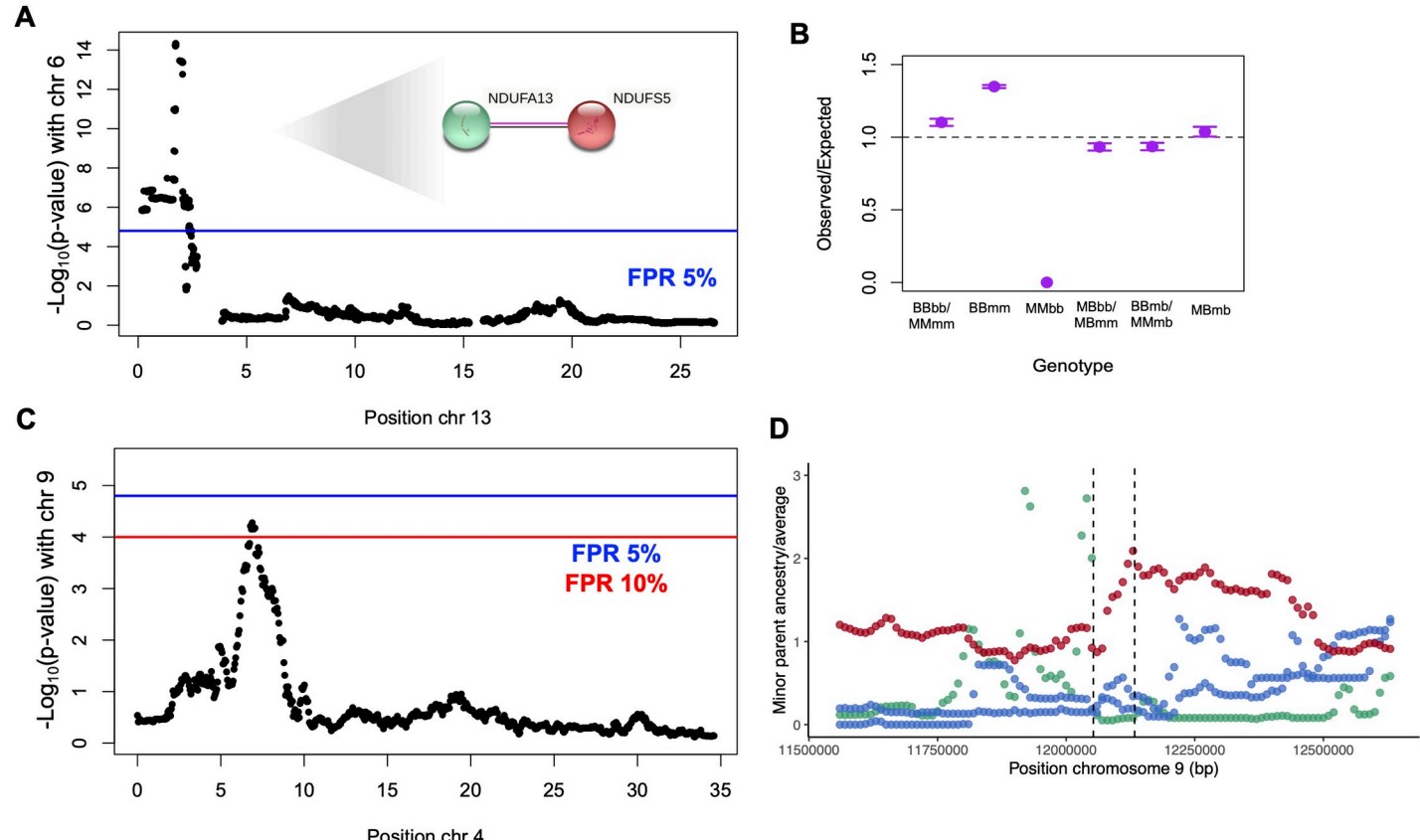

**Fig 4. Evidence for repeated hybrid incompatibilities between *X. birchmanni* × *X. malinche* and *X. birchmanni* × *X. cortezi* hybrids. A.** Interacting locus of the shared chromosome 6 ancestry desert identified in a genome-wide scan. This region was identified scanning for deviations from expected two locus genotypes using a $\chi^2$ test in $F_2$ hybrids, based on observed genotypes at the shared chromosome 6 desert and genotypes at other loci throughout the genome. Genome-wide significance threshold (FPR 5%; blue line) was determined using permutations, see Methods for details. Inset shows STRING network with experimentally verified interactions between *ndufa13* (contained in the chromosome 6 region) and *ndufs5* (contained in the chromosome 13 region). **B.** Ratio of expected to observed two-locus genotype combinations in $F_2$ hybrids between *X. birchmanni* × *X. malinche* at the shared chromosome 6 desert and associated chromosome 13 region. Capital letters indicate genotype at chromosome 6 and lowercase letters indicate genotype at the chromosome 13 region. For visualization, certain genotype combinations are collapsed (i.e. homozygous parental–BBbb & MMmm; heterozygous locus 1, homozygous locus 2 –MBbb & MBmm). Note that several genotype combinations were expected to be found at low frequency due to low *X. birchmanni* ancestry on chromosome 13 (MMbb, MBbb, BBbb). **C.** A second interaction between the minor parent desert on chromosome 9 and a region on chromosome 4 is detected at a relaxed FPR of 10%. **D.** This minor parent desert is shared between *X. birchmanni* × *X. cortezi* hybrid populations and the Acuapa *X. birchmanni* × *X. malinche* hybrid population and occurs on chromosome 9 at 12.1 Mb. Points represent minor parent ancestry calculated in 10 kb non-overlapping windows divided by the proportion of the genome derived from the minor parent species. Red–Tlatemaco (*malinche* × *birchmanni*), Blue–Acuapa and Aguazarca (*birchmanni* × *malinche*), Green–Santa Cruz (*birchmanni* × *cortezi*).

high expression network centrality compared to matched null datasets. We did not find clear evidence that the interaction networks of genes in minor parent deserts (or islands) deviated from the genome-wide background (Text J in S1 File).

**Table 1. Nucleotide (Nt) and amino acid (AA) changes between species pairs in interacting genes between the chromosome 6 and chromosome 13 regions.** Ratios of rates of nonsynonymous to synonymous substitutions (dN/dS) inferred by codeml [60] are also shown. Comparisons where dN/dS is marked as > 99 are due to all nucleotide changes being nonsynonymous.

| | *ndufs5* | | | *ndufa13* | | |
|---|---|---|---|---|---|---|
| **Species Comparison** | **dN/dS** | **Nt changes** | **AA changes** | **dN/dS** | **Nt changes** | **AA changes** |
| *X. birchmanni—X. cortezi* | > 99 | 5 | 5 | > 99 | 3 | 3 |
| *X. birchmanni—X. malinche* | > 99 | 5 | 5 | 1.2 | 4 | 3 |
| *X. cortezi–X. malinche* | 0 | 0 | 0 | 0.001 | 1 | 0 |

## Discussion

We now know that species across the tree of life have hybridized with their close relatives and that many species derive substantial proportions of their genomes from past hybridization events. Based on these observations, it is likely that hybridization plays an important role in the evolution of genomes and species, but we still understand relatively little about how this process unfolds. There are major open questions about which mechanisms shape retention and loss of minor parent ancestry after hybridization and how repeatable this process is across species.

While the predictability of evolution has been a classic question in evolutionary biology for decades [61], such questions have been less central to the hybridization literature [6, 10, 62]. With the expanding number of known hybridization events between closely related species, we are now poised to tackle these questions. One general principle that has emerged is that ancestry from the minor parent species, or the species that contributed less to the genome of hybrids, is more frequently purged by selection after hybridization [6]. This purging can be explained by several mechanisms of selection on hybrids, including selection against hybrid incompatibilities [19, 25, 28, 29, 34], and tends to be stronger in regions of the genome with many linked coding or conserved basepairs [8, 12, 34]. We also uncover these patterns in the *X. birchmanni* × *X. cortezi* hybrid populations studied here, showing a substantial effect of local recombination rate and linked coding and conserved basepairs on minor parent ancestry in hybrids (Fig 2). This represents at least the sixth system in which these effects have been detected [8, 12, 34, 35, 37, 39], suggesting that these patterns of selection on minor parent ancestry are predictable outcomes of the process of genome stabilization after hybridization.

Here, we dramatically expand our understanding of how repeatable the outcomes of hybridization are by comparing patterns of local ancestry in hybridization events between *X. birchmanni* × *X. cortezi* and *X. birchmanni* × *X. malinche* species pairs. This allows us to evaluate, for the first time, some of the factors driving repeatability in the outcomes of hybridization across related species pairs with similar histories of admixture. Both hybridization events began recently (in the last ~80–150 generations [20, 34]) but occur in different geographic regions between distinct species pairs (Fig 1). Because we find that broad scale factors like genome structure are important in generating local ancestry variation in both species pairs, this is expected to drive some concordance in local ancestry across the two species pairs. Indeed, we detect repeatability in where in the genome minor parent ancestry is retained or purged across pairs of *X. birchmanni* × *X. cortezi* and *X. birchmanni* × *X. malinche* hybrid populations (Fig 2 and S10 Table).

Beyond these broad patterns generated by shared genetic architecture, little is known about whether the same genes or regions of the genome are intolerant of hybridization in related species pairs. Intriguingly, we still observe correlations in local ancestry across the two types of hybrid populations after accounting for shared genomic architecture (S12 Table). This indicates that although shared genome structure plays a role in some of the repeatability in local ancestry we observe, other factors are likely to contribute, such as shared sites under selection across the two types of hybridization events. Indeed, when we searched for local signals of shared ancestry across the two hybrid population types, we found evidence that some of the same regions of the genome are under selection in hybrid populations of both species pairs. Compared to expected overlap by chance, *X. birchmanni* × *X. cortezi* and *X. birchmanni* × *X. malinche* hybrid populations are >10X enriched in both shared minor parent deserts and islands.

What factors contribute to this high repeatability of sites under selection across the two types of hybrid populations? From first principles, possible mechanisms driving repeatable

local ancestry include shared hybrid incompatibilities or shared regions of adaptive introgression. Given that the genetic architecture of hybrid incompatibilities has been well-studied in *X. birchmanni* × *X. malinche* hybrids, we can directly ask about the role of such sites in contributing to repeatable ancestry across the two types of hybrid populations. In regions identified as likely hybrid incompatibilities in $F_2$ hybrids between *X. birchmanni* × *X. malinche*, minor parent ancestry was on average lower near these sites in *X. birchmanni* × *X. cortezi* hybrid populations. Only two of these regions overlap with the minor parent deserts discussed above, suggesting that they represent largely complementary signals of shared sites under selection in replicate *X. birchmanni* × *X. cortezi* and *X. birchmanni* × *X. malinche* hybrid populations.

Further comparisons across hybrid population types reveal at least one and possibly multiple shared hybrid incompatibilities between *X. birchmanni* × *X. cortezi* and *X. birchmanni* × *X. malinche*. We identified a single region on chromosome 6 that was a segregation distorter in early generation hybrids and depleted in minor parent ancestry in all five replicate *X. birchmanni* × *X. cortezi* and *X. birchmanni* × *X. malinche* hybrid populations. We found that this region on chromosome 6 interacts with a locus on chromosome 13 (Fig 4). Based on gene network analysis, we identified two mitonuclear genes in the chromosome 6 and chromosome 13 regions that physically interact to form part of mitochondrial complex I. Intriguingly, all hybrid populations studied here of both *X. birchmanni* × *X. cortezi* and *X. birchmanni* × *X. malinche* are fixed for mitochondrial haplotypes from the major parent species, and crosses only included individuals with *X. malinche* mitochondria. This raises the possibility that the incompatibility may be driven by complex interactions between the mitochondrial genome and its interacting nuclear counterparts. Indeed, we find evidence that this is the case in *X. birchmanni* × *X. malinche* hybrids in a companion study [59]. Similar analyses of other regions suggest that there are multiple shared hybrid incompatibilities between *X. birchmanni* × *X. cortezi* and *X. birchmanni* × *X. malinche* associated with shared minor parent ancestry deserts (Fig 4).

While it is clear that hybrid incompatibilities play a role in driving some of the shared minor parent deserts across the two types of hybrid populations, other mechanisms of selection are likely important in generating shared minor parent islands (although we note that minor parent islands are less strongly enriched compared to expectations by chance). In some cases, they may reflect cases where the minor parent haplotype is globally beneficial (i.e. adaptive introgression). For example, *X. birchmanni* is the minor parent in both the Santa Cruz and Tlatemaco populations, hinting that *X. birchmanni* haplotypes may be globally favored within the four shared minor parent islands identified in these two populations. In other pairs of populations, however, when the same regions are enriched in minor parent ancestry this means that different ancestries are favored in the two populations (i.e. in the seven islands shared between Santa Cruz and Aguazarca, *X. birchmanni* ancestry is overrepresented in the former and *X. malinche* in the latter). We speculate that some minor parent islands could correlate with regions under balancing selection such that minor parent ancestry is generally advantageous or other mechanisms that can favor minor parent ancestry such as masking of weakly deleterious alleles [63]. Notably, three shared minor parent islands occur in inversions that differentiate species, conflicting with the prevailing wisdom that inversions should resist introgression (S18 Fig; see [39, 55, 56] for similar findings).

The shared hybrid incompatibility we identify here in *X. birchmanni* × *X. malinche* and *X. birchmanni* × *X. cortezi* hybrids is one of the first described cases in any species group. However, this finding should not be surprising since some hybrid incompatibilities are predicted to be shared between related species from first principles. Hybrid incompatibilities are typically described as interactions between derived mutations in each species but incompatibilities can also arise between the ancestral and derived genotypes when multiple derived mutations have

accumulated along one lineage (S25 Fig; [43]). This type of ancestral-derived incompatibility may be more likely in cases where there are biased patterns of lineage specific substitutions in particular gene classes, as appears to be the case for the *X. birchmanni* mitochondrially interacting proteins discussed above (Table 1). Our findings add to a handful of known cases where the same or similar genetic interactions are under selection in hybridization events across multiple species pairs [21, 23, 64, 65]. Whether this phenomenon is widespread awaits progress mapping additional hybrid incompatibilities in related species pairs [6]. The comparative approach used here is a powerful strategy for evaluating this question on a genome-wide scale.

Together, our results highlight predictable outcomes of selection after hybridization. We find that the outcomes of hybridization are repeatable across closely related species pairs, and are driven in part by broad scale factors such as shared genetic architecture. Beyond shared genomic architecture, we find that diverse mechanisms contribute to repeatability in local ancestry, including shared hybrid incompatibilities between species and shared sites where minor parent ancestry is favored. This parallelism at both the broad and local scale is an important first step towards untangling the drivers of local variation in ancestry in hybrid genomes.

## Methods

### Ethics statement

All animal use followed protocols that were approved by Stanford IACUC (Stanford APLAC protocol number 33071). We are grateful to the Mexican federal government for permission to collect samples (permit number: PFF/DGOPA-064/20).

### Sample collection

Wild fish were caught using baited minnow traps in the states of Hidalgo and San Luis Potosí, Mexico [45]. Individuals from populations that contain a mixture of pure *X. birchmanni* individuals and *X. birchmanni* × *X. cortezi* hybrids were sampled from two distinct collection sites occurring in separate tributaries of the Río Santa Cruz in northern Hidalgo: Huextetitla (21˚ 9'43.82"N 98˚33'27.19"W) and Santa Cruz (21˚9'27.63"N 98˚31'13.79"W). We excluded pure *X. birchmanni* individuals based on genome-wide ancestry, resulting in 254 hybrid individuals included in this study. Ninety-eight of these individuals were originally sequenced as part of a previous study documenting the hybridization event between *X. cortezi* and *X. birchmanni*, and the remaining individuals were collected from the Santa Cruz hybrid population in July of 2020. Collected fish were anesthetized in a 100 mg/mL buffered solution of MS-222 and water (Stanford APLAC protocol #33071). Fish were then photographed and a small fin clip was taken from each individual's caudal fin. Fin clips were preserved in 95% ethanol for later DNA extraction.

### Tn5 libraries for low-coverage whole genome sequencing of *X. birchmanni* × *X. cortezi* hybrids

We prepared libraries for low-coverage whole genome sequencing from DNA extracted from fin clips collected from fish caught at the Huextetitla and Santa Cruz populations as described in Powell et al. [45]. Briefly, we extracted DNA from fin tissue using the Agencourt DNAdvance kit (Beckman Coulter, Brea, California) using half the recommended reaction volume but otherwise as specified by the manufacturer. We quantified isolated DNA with a TECAN Infinite M1000 microplate reader (Tecan Trading AG, Switzerland). We then prepared tagmentation-based whole genome libraries for low coverage sequencing by enzymatically

shearing DNA diluted to approximately 2.5 ng/ul using the Illumina Tagment DNA TDE1 Enzyme and Buffer Kits (Illumina, San Diego, CA) at 55˚C for 5 minutes. We amplified the sheared DNA in dually-indexed PCR reactions for 12 cycles. We purified pooled amplified PCR reactions with 18% SPRI magnetic beads. Library concentrations were measured with a Qubit fluorometer (Thermo Scientific, Wilmington, DE) and library size distributions were measured using Agilent 4200 Tapestation (Agilent, Santa Clara, CA). Libraries were sent for sequencing on an Illumina HiSeq 4000 at Admera Health Services, South Plainfield, NJ.

### Local ancestry inference in *X. birchmanni* × *X. cortezi* hybrids

We previously developed and validated local ancestry inference methods for *X. birchmanni* × *X. cortezi* hybrids using a combination of demographic inference, simulations, analysis of parental individuals, and analysis of a subset of the hybrid individuals described here [45]. Briefly, we used panels of parental individuals (N = 37 for *X. cortezi* and N = 55 for *X. birchmanni*) to identify 1.1 million ancestry informative sites genome wide that were nearly fixed between the two parental species (90% frequency difference or greater), yielding approximately 3 ancestry informative sites per 2 kb. Performance was tested on pure parental individuals not included in the reference panel and in a series of simulations [45]. We found that across scenarios tested, accuracy in local ancestry inference was remarkably high (≤0.3% errors per ancestry informative site in all tested scenarios), presumably due to the high density of ancestry informative markers.

We applied a hidden Markov model (HMM) to infer local ancestry based on low-coverage whole genome sequencing data collected from Huextetitla and Santa Cruz hybrids, using the pipeline described above [45, 66]. Average coverage across individuals was 1.1X and the minimum coverage cutoff used in this study was 0.15X. We set priors for the number of generations since initial admixture to 150 and the admixture proportion to 85% *X. cortezi* based on the results of previous work [45]. We set the error rate for the HMM to 0.02 based on empirically inferred error and cross-contamination rates for our library preparation protocol [66]. After running the HMM and excluding ancestry informative sites that were not covered in any individual, we recovered posterior probabilities for each ancestry state at 1,050,362 informative sites genome wide in 254 individuals.

For many of our analyses, it was convenient to convert these posterior probabilities for a given ancestry state to hard-calls. We used a posterior probability threshold of 0.9 or greater to assign a site to a given ancestry state (homozygous *X. birchmanni*, heterozygous, or homozygous *X. cortezi*). Sites with lower than 0.9 posterior probability were masked, as were sites that were present in fewer than 25% of individuals. Minor parent ancestry was summarized across individuals by averaging the ancestry hard-calls at each site across a range of non-overlapping window sizes (e.g. 10 kb– 1 Mb; 0.1–1 cM).

Historical recombination events in admixed populations are detectable as ancestry transitions in present day individuals [67]. We were also able to use this data to infer the approximate locations of crossover events in a hybrid ancestor that resulted in a transition between ancestry states. We identified the interval over which the posterior probability changes from 0.9 for one ancestry state to 0.9 in support of a different ancestry state. We excluded markers that were not in Hardy-Weinberg equilibrium at a Bonferroni corrected p-value of 0.1, since errors in genotyping and ancestry inference could generate false ancestry switches (excluded sites in this analysis: 84,366–8% of ancestry informative markers). This resulted in the identification of 168,858 ancestry transitions genome-wide across the two populations with a median resolution of 22 kb (~29 per individual per chromosome).

The density of ancestry informative sites in swordtails varies with features of interest such as the local recombination rate and the number of linked functional basepairs [34], likely due

to processes such as background selection within the parental lineages. As a result, we must be cautious of technical biases that could arise due to the distribution of ancestry informative sites that could mislead us into inferring biological differences when there are in fact none. To address this, we thinned the data with respect to the density of ancestry informative sites and re-inferred local ancestry as described above. On average, an ancestry informative site between *X. birchmanni × X. cortezi* occurs once every 395 basepairs. For the ~50% of the genome where ancestry informative sites occurred more frequently, we thinned informative sites to retain only one per 395 basepairs. We inferred ancestry using this thinned set of informative sites (N = 689,983) and performed all of the analyses described on both thinned and unthinned datasets to ask if any results qualitatively change as a result of the input dataset. Results for the thinned datasets are reported in Text K in S1 File.

## Local ancestry inference in *X. birchmanni × X. malinche* hybrids

For the replicate *X. birchmanni × X. malinche* hybrid populations analyzed in our study, we took advantage of data collected from previous studies of these replicate populations [34, 57]. As previously described, we inferred local ancestry with an HMM and used population specific priors for the admixture proportions and time since initial admixture [23, 34]. We limited our analysis to individuals collected over a single sampling year. Briefly, we included 95 individuals collected from Tlatemaco in 2013 (resulting in 629,584 ancestry informative sites), 97 individuals collected from Acuapa in 2018 (613,171 sites), and 51 individuals collected in 2016 from Aguazarca (628,811 sites). Data deposited in the Dryad repository: doi:10.5061/dryad.4mw6m90bp [68].

Previous work has explored the accuracy of these local ancestry inference approaches in *X. birchmanni × X. malinche* hybrids in great detail [23, 66]. Briefly, initial definition of ancestry informative markers relied on 183 parental individuals [23], ancestry informative sites that were not fixed or nearly fixed between the two parental species were excluded (<98% frequency difference). Performance was evaluated in simulations, on pure parental individuals not used in the definition of ancestry informative sites and on early generation $F_1$ and $F_2$ hybrids where switches in ancestry over short genetic distances can confidently be inferred to be errors. Based on these analyses we estimated error rates to be $\leq 0.2\%$ errors per ancestry informative site in all tested scenarios [23, 66]. Thus, given high expected accuracy of ancestry inference in both *X. birchmanni × X. cortezi* and *X. birchmanni × X. malinche* hybrids, we do not expect that errors will drive to the global or local patterns that we observe in the empirical data.

## Comparing minor parent ancestry as a function of local genetic architecture

For the *X. birchmanni × X. cortezi* hybrid populations, we wanted to evaluate whether minor parent ancestry was correlated with recombination rate, as had previously been described in other species [34, 35, 39]. We summarized average ancestry in non-overlapping 50 kb, 100 kb, and 250 kb windows for each population. We previously created a linkage disequilibrium (LD) based recombination map for *X. birchmanni* [34] and calculated recombination rate based on this map over these same spatial scales. Available data suggests that recombination maps are likely to be conserved between *X. birchmanni* and *X. cortezi* (Text D in S1 File; [34]), and between parental species and hybrids [34].

Because the local recombination rate covaries with other genomic features in swordtails [34], we also calculated the number of coding (or conserved) basepairs in each window so that we could account for this in our analysis. We performed these calculations using annotations

developed for the *X. birchmanni* genome [23]. We then used a Spearman's partial correlation in R to determine if ancestry was correlated with recombination rate, after accounting for the number of coding and conserved basepairs in each window. To aid in interpreting these empirical results, we also analyzed simulations that were performed under the inferred demographic history of *X. birchmanni* × *X. cortezi* and *X. birchmanni* × *X. malinche* hybrid populations, with and without selection (Texts C and F in S1 File).

After evaluating the relationship between recombination rate and ancestry in *X. birchmanni* × *X. cortezi* populations, we were also interested in the relationship between minor parent ancestry and the locations of coding and conserved basepairs along the genome. To control for recombination rate, which is positively correlated with coding and conserved bases in swordtails [34], we summarized average ancestry in 0.25 cM non-overlapping windows. As noted above, we used the gene annotation file for *X. birchmanni* and a set of regions conserved in percomorph fish that we had previously identified using phastCons. Comparisons of alignments across species indicate that the locations of these elements are largely conserved across *X. birchmanni*, *X. malinche*, and *X. cortezi* (Text D in S1 File). We counted the number of coding or conserved basepairs in each window. Using this approach, we evaluated the relationship between ancestry and the number of linked coding or conserved basepairs with a Spearman's correlation (and partial correlation) in R. We chose to present results for 0.25 cM windows in the main text because 50% of minor parent ancestry tracts in Santa Cruz are 0.25 cMs in length or longer (75% are 0.1 cMs in length or greater). However, we evaluated these relationships across a range of non-overlapping window sizes (0.1–1 cM) and found the results were qualitatively similar (S4 Table).

Because power to accurately infer ancestry may be correlated with the number of nearby conserved basepairs, we also repeated these analyses using ancestry probabilities generated from a set of ancestry informative sites thinned to reduce power differences between different regions of the genome (see above, *Local ancestry inference in X. birchmanni × X. cortezi hybrids*). We found that our results were qualitatively unchanged (S8 Table). We also repeated analyses of the relationship between the number of conserved and coding basepairs and minor parent ancestry, excluding windows that fell in the lower or upper 25% quantile of the number of ancestry informative sites. We found that excluding these windows, where we expect to have particularly low or high power to infer ancestry, did not impact our results (S9 Table).

For analyses including recombination rate, we repeated tests using an LD-based map that we inferred using the same methods after thinning SNPs in regions of the genome with high SNP density in the *X. birchmanni* population sample it was originally generated with [34]. Because our ability to reliably estimate recombination rate depends on the density of SNPs, we expect our power to estimate recombination rate to vary along the genome. If power to estimate recombination rate varies with other features of interest, such as density of coding or conserved basepairs, this could mislead us into interpreting variation in power as variation in signal. To evaluate this possibility, we calculated the median distance between SNPs in our dataset, which was 184 basepairs. We thinned the SNPs input into LDhelmet so that when SNPs were denser than one every 184 basepairs, a single SNP was retained per 184 basepairs. We then re-inferred recombination rates using LDhelmet as described previously [34]. We repeated all analyses described above that incorporate recombination rate with this thinned LD map and found the results qualitatively unchanged (Text K in S1 File).

As a further validation of the relationship between minor parent ancestry and recombination rate, we repeated analyses with a lower-resolution crossover-based map generated from the 943 $F_2$ *X. birchmanni* x *X. malinche* hybrids. We observed a total of 34,939 ancestry transitions genome-wide in our dataset, or an average of 1.5 per individual per chromosome. We summarized the number of observed ancestry transitions, reflecting crossovers in the $F_1$

parents, in windows. On average we observed only ~5 crossovers per 100 kb, so we chose to summarize recombination rate in large windows of 1 Mb and 5 Mb. We repeated analyses of the relationship between recombination rate and minor parent ancestry using this lower resolution crossover map and found that our results were consistent with those recovered using the LD-based maps (S3 Table).

Finally, we tested the robustness of our results in several additional ways to control for a number of potential technical artifacts (see full description in Text K in S1 File and S2 Table). Since unexpectedly short ancestry tracts may represent errors, we repeated our analyses masking minor parent ancestry tracts that were shorter than 0.004 cM and major parent ancestry tracts that were shorter than 0.035 cM (tract length expectations given population history were derived from [69]). To control for potential autocorrelation in nearby windows we calculated correlations between minor parent ancestry and features of interest when we retained one window per 500 kb (approximately the scale of admixture LD in these hybrid populations; [34, 45]). To control for the fact that recombination rates in hybrids are expected to differ within species-specific structural rearrangements, we identified and removed windows spanning known inversions and repeated our analyses.

## Repeatability of local ancestry between cross types

To evaluate correlations in minor parent ancestry across *X. birchmanni* × *X. cortezi* and *X. birchmanni* × *X. malinche* populations we used summaries of observed ancestry in non-overlapping physical (50–250 kb) and genetic windows (0.1–1 cM), as described above. We first calculated pairwise correlations between populations in minor parent ancestry without accounting for other features (Figs 2 and S10 Table). We then repeated these calculations using partial correlation tests in physical windows (50–250 kb) that included the recombination rate, the number of linked coding, and conserved basepairs as covariates (S12 Table) and partial correlation tests in genetic windows (0.1–1 cM) that included the number of linked coding and conserved basepairs as covariates (S13 Table). This allowed us to infer both whether broad-scale correlations in local ancestry could be detected across populations and whether these correlations could be largely explained by shared genetic architecture (i.e. locations of coding and conserved basepairs and local recombination rate). We also explored this question using simulations. We performed simulations under the inferred demographic histories for *X. birchmanni* × *X. cortezi* and *X. birchmanni* × *X. malinche* hybrid populations with and without selection (Texts C and F in S1 File) and evaluated expected cross-population correlations in ancestry under different scenarios.

## Identification of minor parent ancestry "deserts" and "islands"

Beyond shared correlations as a result of shared broad-scale architecture of the genome, we were interested if we could identify shared minor parent ancestry "deserts," where minor parent ancestry was unusually low, and shared minor parent ancestry "islands, where minor parent ancestry was unusually high, that were shared across *X. birchmanni* × *X. cortezi* and *X. birchmanni* × *X. malinche* hybrid populations.

To do so we identified regions of especially high and low minor parent ancestry in Santa Cruz, the *X. birchmanni* × *X. cortezi* hybrid population for which we have the largest dataset (for results of the same analysis using the Huextetitla populations see Text L in S1 File). We first identified ancestry informative sites where the minor or major parent ancestry fell in the lower 2.5% tail of genome-wide ancestry. We expanded out in the 5' and 3' direction from the focal ancestry informative site until we reached a site on each edge that exceeded the 5% ancestry quantile, and treated this interval as the boundary of the minor parent ancestry desert or

island. To control for possible technical artifacts, we then evaluated ancestry in these regions in 0.05 cM windows. We identified the window that contained the midpoint of the islands or desert and checked that this entire window fell in the 10% minor or major ancestry tail, contained at least 10 ancestry informative sites, and was more than 10 kb in length. We also merged any detected regions that were closer together than 50 kb (deserts N = 54, islands N = 14) and removed any regions shorter than 10 kb (deserts N = 1). We also filtered regions where inferred ancestry varied substantially between inference with unthinned and thinned ancestry informative markers (S13 Fig). Specifically, we filtered islands where inferred ancestry differed by more than 10% between thinned and unthinned data (N = 58, based on 0.05 cM windows). This resulted in the identification of 81 minor parent ancestry deserts with an average length of 317 kb, and 76 islands with an average length of 242 kb in the Santa Cruz population.

We next wanted to compare these regions across populations. Because each population has a distinct recombination history and distribution of ancestry tract sizes, we took the midpoint of the deserts and islands identified in the Santa Cruz hybrid population and overlapped these regions with average ancestry in other populations, calculated at a fine spatial scale (0.05 cM, or on average 11 kb). We then determined if these regions had unusually high or low minor parent ancestry in other populations (defined as falling in the 10% minor or major ancestry tail in another population). We identified several shared islands and deserts between Santa Cruz and *X. birchmanni* × *X. malinche* populations (Figs 2 and 3). Simulations suggest that this two-step approach has excellent power to detect shared sites under selection, even when selection coefficients are relatively weak (Text G in S1 File).

Next, we asked whether the number of shared minor parent islands and deserts across *X. birchmanni* × *X. malinche* and *X. birchmanni* × *X. cortezi* hybrid populations exceeded what would be expected by chance. To investigate this, we used the same 0.05 cM windows we used above to determine whether minor parent ancestry deserts and islands were shared. We permuted the average ancestry values in *X. birchmanni* × *X. malinche* populations across windows and asked whether any windows that were major or minor parent ancestry outliers in the permuted data overlapped with the ancestry deserts identified in Santa Cruz (defined as falling in the 10% tail of major or minor parent ancestry, as in the real data). We repeated this procedure 1000 times. Based on these permutations, we found that few minor (or major) parent ancestry outliers in *X. birchmanni* × *X. malinche* hybrid populations overlap minor parent deserts (or islands) identified in the Santa Cruz hybrid population by chance (Fig 3A). For pairs of populations, the expected false positive rate across all simulations was less than 10% (Fig 3A). When we instead asked about the probability that any minor parent desert or island identified in Santa Cruz overlapped an ancestry outlier in simulations, the false positive rate was higher (~20%).

Our interest in performing permutations was to determine how much overlap in minor parent ancestry deserts and islands is expected by chance, and how much we can attribute to shared sources of selection on hybrids. While our permutations above suggest that little overlap is expected by chance, one challenge of this type of analysis is that permuting the data disrupts all correlations in ancestry across hybrid populations. This means that it will also disrupt correlations driven by interactions between selection and features of the genome such as the local recombination rate and coding/conserved basepair density. Since we are primarily interested in minor parent deserts and islands driven by sites under shared selection rather than windows that happen to be in low recombination rate or high functional density regions of the genome, we attempted to control for this. First, our permutations use genetic windows, which account in part for the relationship between minor parent ancestry and recombination rate variation. To evaluate whether minor parent ancestry deserts and islands were likely driven by

functional basepair density, we compared the windows identified in our analysis to the genomic background. We describe these methods in detail in Text H in S1 File and the results of these analyses are presented in S14–S16 Figs.

Given the importance of this analysis for the conclusions of our manuscript, we also took two additional approaches to determine the expected false positive rates for desert and island regions. In the first, we shuffled ancestry in blocks along the genome to preserve local ancestry structure during permutations but otherwise evaluated false positive rates as described above. We found the results of these permutations to be similar, but with a somewhat higher false positive rate for minor parent deserts and a substantially higher false positive rate for minor parent islands (Text H in S1 File and S12 Fig). In the second, we analyzed expected false positive rates for each identified desert and island individually. Briefly, for each minor parent desert or island we generated windows of matched cM length, and selected all windows in the genome that had a similar content of coding basepairs. We next used permutations to determine the expected overlap by chance of regions of extreme ancestry within these matched sets (Text H in S1 File).

## Identifying loci under selection using controlled crosses and natural populations

Hybridization between *X. birchmanni* × *X. malinche* has been studied for more than a decade. As such, we have generated controlled crosses between *X. birchmanni* × *X. malinche* that are not yet feasible between *X. birchmanni* × *X. cortezi*. We have also mapped a number of hybrid incompatibilities that occur between the two species, including a gene interaction that causes melanoma in hybrids [23] and dozens of pairs of loci that show signatures consistent with selection against DMIs in natural hybrid populations [20, 49].

We also recently generated a large mapping panel of $F_2$ hybrids between *X. birchmanni* × *X. malinche*, which we used for QTL mapping of male sexually selected traits [57]. Here, we reanalyzed this mapping panel, doubling our sample size by including females, to identify strong segregation distorters. Segregation distortion can be generated by selfish genetic elements but in hybrids segregation distortion is often a hallmark of regions of the genome that impact hybrid viability or fertility [31]. Since segregation distorters that we have power to detect in our mapping panel are likely to have a strong impact on viability or fertility of *X. birchmanni* × *X. malinche*, they also represent loci that we were likely unable to map previously since our initial work on hybrid incompatibilities relied on segregating variation in *X. birchmanni* × *X. malinche* hybrid populations [20, 23]. Briefly, we generated $F_1$ hybrids between *X. malinche* females and *X. birchmanni* males, and produced $F_2$s by crossing this $F_1$ generation [57]. The reverse cross direction is rarely successful [59]. We inferred local ancestry as described previously for this cross [57, 66]. As expected from the cross design, average ancestry across ancestry informative sites in our dataset of 943 males and females indicated that on average 50% of the genome was derived from each parental species. Our previously published dataset for QTL mapping included 568 males [57].

In reanalyzing this data, we next turned to identifying regions of the genome with unusually high *X. birchmanni* or *X. malinche* ancestry, consistent with loci having a major impact on viability or fertility in hybrids (or generating segregation distortion for other reasons). We first thinned the $F_2$ data to retain one ancestry informative marker per 50 kb for computational efficiency. We then used a binomial test to identify loci that deviated from the expected segregation ratio of 50% *X. birchmanni* and 50% *X. malinche* ancestry. We used a p-value threshold of $p < 5^{-4}$, approximately corresponding to the 5% tail of ancestry outliers in our dataset. We note that several of these regions were robust to Bonferroni correction for the number of markers

tested (N = 20,011; p$<5^{-6}$ corresponding to FDR of 10%). We consider this analysis conservative due to the extent of ancestry linkage disequilibrium in $F_2$ hybrids. After identifying regions of the genome with evidence for segregation distortion, we visualized ancestry in each region to ensure that the pattern was consistent with an extended region of segregation distortion rather than a short interval as might be expected from e.g. genotyping error (see S20 Fig).

To explore the strength of selection that would be required to generate segregation distortion at the threshold we impose on the real data, we performed simple simulations. We simulated genotypes formed from 50–50 mixtures of the two parental gametes. Next, we randomly drew a selection coefficient from 0–1 against one of the homozygous genotypes and a dominance coefficient (0–1). We implemented selection on this pool of genotypes, treating 1-*s* (or 1-*hs*) as the probability of survival. We drew 943 individuals from the surviving genotypes and calculated average ancestry at the simulated sites. Simulations where ancestry fell in the 5% tail of the real data were treated as accepted simulations; we repeated this procedure 10,000 times. Based on the distribution of accepted simulations (S26 Fig), we predict that very strong selection (*s* > 0.25) is required to generate ancestry patterns as extreme as the loci we focus on in our $F_2$ dataset. Distributions of the dominance coefficient from accepted simulations mirrored the prior distribution.

## Role of loci under selection in ancestry variation in natural populations

One of our major questions is whether loci under selection are shared not just across hybrid populations of the same cross type but between hybrid populations of different cross types. To tackle this question we took advantage of what is known about selection in the *X. birchmanni* × *X. malinche* system, using the segregation distortion regions identified above, and more weakly selected putative DMIs that have been previously identified in natural hybrid populations between *X. birchmanni* × *X. malinche* [20, 49]. Both sets of loci are associated with lower minor parent ancestry in *X. birchmanni* × *X. malinche* hybrid populations [34], as expected by theory.

We asked if these loci were also associated with lower minor parent ancestry in *X. birchmanni* × *X. cortezi* hybrid populations, and the degree to which they coincided with ancestry deserts that are shared across *X. birchmanni* × *X. cortezi* and *X. birchmanni* × *X. malinche* hybrid populations. We summarized ancestry in 10 kb non-overlapping windows and used bedtools [70] to determine the distance of each window to the closest selected site in each of the two datasets and plotted average major parent ancestry as a function of that distance (S27 Fig).

## Identifying potential partners of shared ancestry desert and segregation distortion region

One of the regions that we identified as a significant segregation distorter on chromosome 6 in $F_2$ hybrids between *X. birchmanni* × *X. malinche* was also a shared minor ancestry desert, and had unusually low minor parent ancestry across all surveyed *X. birchmanni* × *X. malinche* and *X. birchmanni* × *X. cortezi* hybrid populations. The observation that minor parent ancestry is disfavored across populations of diverse ancestry (majority *malinche*, majority *birchmanni*, and majority *cortezi*) indicates that this locus is likely to be involved in a hybrid incompatibility [34], as opposed to being driven by other possible mechanisms of selection on hybrids [6].

Based on these observations, we wanted to identify the interacting partner of the chromosome 6 locus. Given the clear expectations for genotype frequencies in $F_2$ hybrids and apparent shared selection on this locus across species pairs, we used the $F_2$ data generated between *X. birchmanni* × *X. malinche* to perform this scan. We selected an ancestry informative marker

from the center of the shared ancestry desert on chromosome 6, which falls within the peak segregation distortion region on chromosome 6 identified in $F_2$ hybrids (Fig 3). Next, we performed a scan against all other ancestry informative markers across the genome and calculated $\chi^2$ statistics for deviations from expected two-locus genotype frequencies (with four degrees of freedom). Importantly, we used the observed genotype frequencies across individuals at each of the two loci to generate this expectation, such that we identified loci with deviations from expected two-locus genotype combinations rather than deviations in ancestry at one locus or the other.

To determine the appropriate significance threshold for this scan, we performed simulations. Using the $F_2$ ancestry data described in the previous paragraph, we simulated genotypes at the focal locus based on each individual's genome-wide ancestry and performed the $\chi^2$ test as described above. We repeated this for every locus in the dataset (i.e. over 20,011 markers) and recorded the minimum p-value. We performed this procedure 500 times and took the lower 5% quantile of minimum p-values across simulations. This value served as our false positive rate (FPR) cutoff for analysis of the real data. We describe this approach in more detail in Text I in S1 File. We initially explored an approach using permutations at the focal locus to set the FPR but found that this approach was less conservative; these results are also described in Text I in S1 File. Only one region in the genome (found on chromosome 13) passed the FPR threshold determined by permutation.

We also repeated this scan for interacting loci for all ancestry deserts shared between *X. birchmanni* × *X. cortezi* and *X. birchmanni* × *X. malinche* populations. While we did not identify any interacting loci that passed the 5% genome-wide FPR correction threshold for other regions, we highlight two other cases where interactions are detected with minor parent deserts at a FPR of 10% (Text I in S1 File). Simulations suggest that our power to detect interactions in is relatively low (Text I in S1 File), leaving open the possibility that other minor parent deserts interact with partner loci that we are unable to detect in the current data.

## Pathway analysis of genes falling in a shared hybrid incompatibility

We wanted to ask whether genes in the identified chromosome 6 and chromosome 13 regions might be known to interact from past experimental, fusion, or co-expression data. To evaluate this, we took advantage of the STRING database [58]. We used haltools [71] to liftover coordinates from the *X. birchmanni* reference genome to the *X. maculatus* reference genome for the shared ancestry desert on chromosome 6 and the interacting region on chromosome 13. This allowed us to extract gene IDs for *X. maculatus* in this region, which is included in the STRING database.

We input gene names from both chromosomes (chromosome 6–13 genes; chromosome 13–29 genes) into STRING using the protein batch search function. Two pairs of interacting genes with supporting experimental, fusion, or co-expression data were identified. For one of these pairs, both genes fell within the chromosome 6 region but for the other pair, one gene fell within the chromosome 6 region and the other within the chromosome 13 region.

Given that our power to map interacting regions of shared minor parent deserts is relatively low (Text I in S1 File), we wanted to perform a broad scan to ask whether there was evidence for enrichment of genes that have many protein-protein interactions or dense regulatory networks within shared minor parent ancestry deserts and islands. We thus performed similar analyses for genes found in other shared minor parent ancestry deserts and islands. These analyses included both STRING database searches similar to those described above, and evaluation of enrichment of hub genes from co-expression analyses in minor parent deserts and islands compared to matched null sets (Text J in S1 File).

### Investigating other features associated with ancestry variation

Because theory predicts that many of the changes in genome-wide ancestry that occur in hybrid populations happen rapidly after admixture [29, 38], we were interested in investigating correlations between minor parent ancestry and chromosome-level features. We evaluated whether there were correlations between chromosome length, number of genes per chromosome, and minor parent ancestry using a Spearman's rank correlation test.

We also evaluated minor parent ancestry as a function of structural differences detected between the *X. birchmanni*, *X. malinche*, and *X. cortezi* assemblies. To identify chromosomal inversions between species, we compared de-novo assemblies using MUMmer (mummer-4.0.0beta2) [72]. After aligning the 24 largest linkage groups, which correspond to the 24 *Xiphophorus* chromosomes, we excluded alignments under 5 kb before proceeding with the analysis. We focused our analysis on scanning for large inversions (> 100 kb). To identify the putative inversion breakpoints, we used mummer's show-coords command. By aligning the *X. birchmanni* [23] assembly to *X. cortezi* [45] and to *X. malinche* [23], and comparing to an outgroup sequence (*X. maculatus;* [51]), we determined which inversions were derived in each species. For simplicity, we assumed the inversion configuration detected in the reference genomes was fixed at the species level ($\pi$ in *X. birchmanni*, *X. malinche*, and *X. cortezi* is <0.12%) but we note that some segregating inversions have been detected in *Xiphophorus* [23].

## Supporting information

**S1 File Texts A-L, supplementary information.**
(DOCX)

**S1 Table. Correlations between minor parent ancestry (*X. birchmanni* ancestry) and recombination rate in Santa Cruz and Huextetitla hybrid populations at different non-overlapping window sizes.**
(DOCX)

**S2 Table. Correlations between minor parent ancestry (*X. birchmanni* ancestry) and recombination rate in Santa Cruz and Huextetitla hybrid populations using different approaches to control for variation in power to infer local ancestry or to infer recombination rate.** See Methods and Text K in S1 File for more details on these analyses. AIMs–ancestry informative sites; Rec–recombination. In the mask short tracts analysis, we removed ancestry tracts shorter than 0.004 cM for minor parent ancestry tracts and 0.035 cM for major parent ancestry tracts, based on the reasoning that these unusually short tracts compared to expectations given the age of the hybrid population [34] might represent switch errors. In the thinned physical distance analysis, one window was retained every 500 kb. In the exclude inversions category, we removed chromosomes 21 and 24 which have large inversions between *X. birchmanni* and *X. cortezi*.
(DOCX)

**S3 Table. Correlations between minor parent ancestry (*X. birchmanni* ancestry) and recombination rate in Santa Cruz and Huextetitla hybrid populations using a $F_2$ crossover map to estimate recombination rate.** See Methods for details.
(DOCX)

**S4 Table. Correlations between minor parent ancestry (*X. birchmanni* ancestry) and the number of coding and conserved basepairs in a range of genetic non-overlapping window**

**sizes.** 75% of ancestry tracts found in Santa Cruz and Huextetitla are 0.1 cM or larger.
(DOCX)

**S5 Table. Analysis of the correlation between minor parent ancestry and linked coding and conserved basepairs in 0.25 cM non-overlapping windows, excluding all regions with structural rearrangements between *X. birchmanni*, *X. malinche*, or *X. cortezi*.**
(DOCX)

**S6 Table. Relationship between minor parent ancestry (*X. birchmanni* ancestry) and the number of synonymous and nonsynonymous substitutions found in a non-overlapping window of a given genetic size.** Ancestry was summarized using the results of an HMM run on a set of thinned input ancestry informative sites (see Methods). Minor parent ancestry is reduced in regions with a higher number of both synonymous and nonsynonymous substitutions between *X. birchmanni* and *X. malinche*. This may be driven by a correlation between coding substitutions and regions with a high number of linked coding basepairs (S7 Table).
(DOCX)

**S7 Table. Relationship between minor parent ancestry (*X. birchmanni* ancestry), the number of coding basepairs, and the number of synonymous and nonsynonymous substitutions found in a window of a given genetic size.** Ancestry was summarized using the results of an HMM run on a set of thinned input ancestry informative sites (see Methods).
(DOCX)

**S8 Table. Analysis of the correlation between minor parent ancestry and linked coding and conserved basepairs in 0.25 cM non-overlapping windows.** Ancestry was summarized using the results of an HMM run on a set of thinned input ancestry informative sites (see Methods).
(DOCX)

**S9 Table. Analysis of the correlation between minor parent ancestry and linked conserved basepairs in non-overlapping windows in a filtered dataset.** Here, windows in the lower or upper 25% of ancestry informative sites were dropped to exclude windows where we had especially low or high power to infer ancestry.
(DOCX)

**S10 Table. Cross-population correlations in minor parent ancestry at a range of non-overlapping window sizes, without controlling for recombination rate and coding/conserved basepair covariates.** Santa Cruz and Huextetitla populations are *X. birchmanni* × *X. cortezi* hybrid populations that derive the majority of their genome from *X. cortezi*. Acuapa and Aguazarca are *X. birchmanni* × *X. malinche* hybrid populations that derive the majority of their genome from *X. birchmanni*; Tlatemaco is a *X. birchmanni* × *X. malinche* hybrid population that derives the majority of its genome from *X. malinche*.
(DOCX)

**S11 Table. Cross-population correlations in minor parent ancestry at a range of non-overlapping window sizes, where windows containing rearrangements in any species have been removed, without controlling for recombination rate and coding/conserved basepair covariates.** Santa Cruz and Huextetitla populations are *X. birchmanni* × *X. cortezi* hybrid populations that derive the majority of their genome from *X. cortezi*. Acuapa and Aguazarca are *X. birchmanni* × *X. malinche* hybrid populations that derive the majority of their genome from *X. birchmanni*; Tlatemaco is a *X. birchmanni* × *X. malinche* hybrid population that derives the majority of its genome from *X. malinche*.
(DOCX)

**S12 Table. Cross-population correlations in minor parent ancestry at a range of non-overlapping window sizes, including recombination rate, coding, and conserved basepair covariates.** For each population and window size comparison, Spearman's $\rho$ from cross-population ancestry correlations after accounting for other features is given to the left of each cell. The estimated $\rho$ from the same partial correlation analysis of other features (recombination rate, number of coding basepairs, and number of conserved basepairs) are given one the right. (DOCX)

**S13 Table. Cross-population correlations in minor parent ancestry at a range of genetic non-overlapping window sizes.** (DOCX)

**S14 Table. Minor parent ancestry deserts identified in Santa Cruz number of coding and conserved basepairs in each desert, average recombination rate in the region, and the probability of overlap with ancestry outliers in matched windows in each of the *X. birchmanni* x *X. malinche* hybrid populations.** (XLSX)

**S15 Table. Minor parent ancestry islands identified in Santa Cruz, number of coding and conserved basepairs in each island, average recombination rate in the region, and the probability of overlap with ancestry outliers in matched windows in each of the *X. birchmanni* x *X. malinche* hybrid populations.** (XLSX)

**S16 Table. Shared ancestry resolved deserts genes.** (XLSX)

**S17 Table. Shared ancestry resolved islands genes.** (XLSX)

**S18 Table. GO results minor parent deserts.** (TXT)

**S19 Table. GO results minor parent islands.** (TXT)

**S1 Fig. PSMC results analyzing population history from a single whole-genome sample of *X. malinche* from the sampling site Chicayotla, 18 *X. birchmanni* individuals from the sampling site Coacuilco, and five *X. cortezi* individuals from the sampling site El Nacemiento de Huichihuayán.** For visualization the single *X. malinche* sample was bootstrapped 100 times by resampling with replacement from the genome split into 500 kb segments. Analysis was conducted similarly to [5] with the time segmentation parameter set to 4+25*2+4+6, a $\rho/\theta$ ratio of 2, generation time of two generations per year, and mutation rate of $3.5 \times 10^{-9}$. (TIFF)

**S2 Fig. Genetic diversity in homozygous *X. cortezi* ancestry tracts in individuals from two hybrid populations (Santa Cruz and Huextetitla) and in individuals from nearby allopatric *X. cortezi* parental populations (Las Conchas and Huichihuayán).** (TIFF)

**S3 Fig.** (**Top**) (Left) PCA analysis of the locations of observed ancestry transitions in individuals from the Santa Cruz and Huextetitla hybrid populations. (Right) PCA analysis of pseudohaploid SNP calls derived from low-coverage sequence data of individuals from the Santa Cruz and Huextetitla populations (see Text B in S1 File). Separation along PC2 suggests that the

Santa Cruz and Huextetitla hybrid populations have been somewhat independent in their recent demographic histories. (**Bottom**) Example of heterogeneity in ancestry along chromosome 2 in Huextetitla (blue) and Santa Cruz (pink) *X. birchmanni* × *X. cortezi* hybrid populations, averaged in 10 kb windows). Both populations have regions that are fixed or nearly fixed for both *X. cortezi* and *X. birchmanni* ancestry. Note the strong correlations in local ancestry between the two populations ($\rho = 0.65$, $p{<}10^{-100}$).
(TIFF)

**S4 Fig.** Posterior distributions from Approximate Bayesian Computation (ABC) simulations used to infer the demographic history for a *X. birchmanni* × *X. cortezi* hybrid population Santa Cruz (STAC) (red) and a *X. birchmanni* × *X. malinche* hybrid population Acuapa (ACUA) (pruple). Dot-dashed lines and listed values correspond to the maximum a posteriori or MAP estimate for each distribution. Dotted lines are the 95% quantile range. See Text C in S1 File for complete details of SLiM simulations and rejection sampling approach. Posterior distributions shown here are derived from uniform (or log-uniform) prior distributions of: initial population size, time since admixture (in generations), initial admixture proportion, and migration rate from each parent species. We accepted simulations based on two sets of summary statistics. **A.** Our primary analysis included summary statistics for the median length of minor parent ancestry tracts, average hybrid index, and the coefficient of variation for chromosome-wide ancestry across sampled individuals (ACUA N = 500, STAC N = 500). **B.** In a second analysis we included summary statistics for the median length of minor parent ancestry tracts, average hybrid index, and the coefficient of variation for local ancestry along the chromosome in 250 kb windows. We accepted very few simulations using the second approach (ACUA N = 98, STAC N = 90). We show the accepted simulations here to emphasize that the posterior distributions for most parameters are similar using the two approaches but rely on the inferences made in **A** for almost all analyses. See Text C in S1 File for additional information.
(TIFF)

**S5 Fig.** **A.** Heterogeneity in minor parent ancestry in the real data for a section of chromosome 1 in the Santa Cruz and Acuapa populations. Points show the average minor parent ancestry in 10 kb windows. **B.** Results of one replicate simulation of local ancestry on chromosome 1 for Santa Cruz and Acuapa based on randomly drawn set of demographic parameters from the posterior distribution of ABC simulations that used global variation in admixture proportion as a summary statistic (see Text C in S1 File & S4 Fig for details). **C.** Results of one replicate simulation of local ancestry on chromosome 1 for Santa Cruz and Acuapa based on randomly drawn set of demographic parameters from the posterior distribution of ABC simulations that used local variation in admixture proportion (summarized in 250 kb windows) as a summary statistic. Points show the average minor parent ancestry in 10 kb windows. While these simulations incorporated inferred demographic history for each population they did not implement selection. This results in lower heterogenetity in local ancestry compared to the real data, even when a summary statistic of local variation in admixture proportion was used to accept or reject simulations (see Text C in S1 File & S4 Fig).
(TIFF)

**S6 Fig. GC\*, a measure of GC biased gene conversion, in 5 kb hotspots identified in *X. birchmanni* as well has GC-content matched coldspots. Also shown is GC\* for *X. cortezi* in the same hotspots and matched coldspots identified in *X. birchmanni*.** These patterns suggest an excess of GC-biased gene conversion in hotspots identified in *X. birchmanni* in both species, providing further evidence that the fine scale recombination maps are shared between

species.
(TIFF)

**S7 Fig. In addition to heterogeneity in ancestry within chromosomes, we also observe substantial heterogeneity in ancestry proportion between chromosomes. A.** In both Santa Cruz and Huextetitla this heterogeneity is modestly correlated with chromosome length, suggesting that it may be driven by higher effective recombination rates on shorter chromosomes ($\rho_{\text{Santa Cruz}}$ = 0.43, p $_{\text{Santa Cruz}}$ = 0.036; $\rho_{\text{Huextetitla}}$ = 0.39, $p_{\text{Huextetitla}}$ = 0.058). **B.** The correlation between number of genes per chromosome and chromosome-level ancestry is substantially weaker ($\rho_{\text{Santa Cruz}}$ = 0.12, $p_{\text{Santa Cruz}}$ = 0.58; $\rho_{\text{Huextetitla}}$ = 0.03, $p_{\text{Huextetitla}}$ = 0.90).
(TIFF)

**S8 Fig. No evidence for unusual minor parent ancestry in the Santa Cruz hybrid population in 0.1 cM windows with high nonsynonymous substitution rates between X. cortezi and X. birchmanni (upper 25% genome wide), versus matched windows with no nonsynonymous substitutions but a similar overall coding substitution rate.** For each 0.1 cM window with a high number of nonsynonymous substitutions, we identified a 0.1 cM window with no nonsynonymous substitutions but an overall coding substitution rate (i.e. of synonymous substitutions) within 80–120% of that observed in the focal window. We see no significant differences in the minor parent ancestry distributions of the focal (pink) and matched (blue) windows.
(TIFF)

**S9 Fig. Different models of selection on hybrids generate distinct correlations between minor parent ancestry and recombination rate. A.** In the absence of selection there is no expected relationship between recombination rate and ancestry, and indeed this is what is observed in simulated Santa Cruz and Acuapa populations (single simulation example shown here). Gray points show minor parent ancestry in 250 kb windows, red points and whiskers show the mean and two standard errors of the mean. Inset shows correlation coefficient and p-value for the representative simulation. **B.** In the presence of selection against hybrid incompatibilities, selection drives a positive correlation between minor parent ancestry and recombination rate, regardless of the identity of the major parent species. Shown here are single representative simulations modeling the demographic history of the Santa Cruz and Acuapa populations with incompatibility selection implemented at 20 random pairs of sites throughout the genome. **C.** In the presence of selection against one parent species or the other, we expect to see conflicting directions in the correlation between minor parent ancestry and recombination rate depending on the admixture proportion of the hybrid population. In this set of simulations, a subset of sites derived from the *X. birchmanni* parent were globally disadvantageous, driving different patterns in the majority *X. birchmanni* (Acuapa) and minority *X. birchmanni* (Santa Cruz) hybrid populations. Simulations are described in detail in Text F in S1 File.
(TIFF)

**S10 Fig. Simulations suggest that we do not expect to observe cross-population correlations in local ancestry in the absence of shared sites under selection. A.** Example simulation of local ancestry in Santa Cruz and Acuapa populations modeling inferred demographic history but no selection. Inset shows correlation coefficient and p-value for the pair of simulations. **B.** Example simulation of local ancestry in Santa Cruz and Acuapa populations modeling inferred demographic history and 20 randomly placed shared sites under selection in the two populations. Blue text shows correlation coefficient and p-value for the pair of

simulations.
(TIFF)

**S11 Fig. Schematic of approach used to identify shared minor parent islands and deserts.**
We employed a stepwise approach to identify deserts and islands of minor parent ancestry and
determine if they were shared across populations. Shown here is a hypothetical workflow for
identifying a shared minor parent ancestry island. **A.** We started with identifying AIMs where
average minor parent ancestry at that site exceeded the 97.5% quantile of minor parent ances-
try genome-wide. From that focal site we expanded outward in the 5' and 3' directions to iden-
tify where minor parent ancestry falls below the 95% tail of the genome-wide distribution.
This set the boundary of the focal minor parent island region. We then determined the mid-
point of each region and identified the 0.05 cM window that contains the midpoint. **B.** We
checked that the focal population's minor parent ancestry is greater than the 90% quantile of
minor parent ancestry genome-wide when averaged across this 0.05 cM window. We then
asked if this region is a shared minor parent ancestry outlier in other populations. **C.** Specifi-
cally, we evaluated minor parent ancestry in the midpoint 0.05 cM window in other hybrid
populations. If minor parent ancestry in these populations is exceeded the 90% quantile of that
population's genome wide ancestry distribution we classified that region as a shared minor
parent island. **D.** Example of a minor parent island detected with this work flow. Dashed lines
are the identified boundaries of the island and dotted line is the midpoint. Colored dots corre-
spond to the minor parent ancestry at a given ancestry informative site divided by the genome
wide average for that population. Colored lines indicate the minor parent ancestry for the focal
0.05 cM window divided by the genome wide average for the population.
(TIFF)

**S12 Fig. Shared minor parent deserts between *X. birchmanni* × *X. cortezi* and *X. birch-
manni* × *X. malinche* hybrid populations are enriched compared to expectations by chance
when using a permutation approach to generate null datasets that preserves the structure
of local ancestry correlation in the genome (see Text H in S1 File).** By contrast, minor parent
islands are less enriched compared to null datasets when using this approach. Results shown
here indicate the number of shared minor parent deserts (or islands) between the Santa Cruz
*X. birchmanni* × *X. cortezi* hybrid population and each *X. birchmanni* × *X. malinche* hybrid
population (Acuapa, Aguazarca, and Tlatemaco). Large black circles show the observed num-
ber of shared minor parent deserts or islands. Gray points and boxplots show the expectations
from 130 shuffled datasets tiling the genome (see Text H in S1 File). The column labeled
Xmal/Xbir shows the number of shared deserts or islands between the Santa Cruz *X. birch-
manni* × *X. cortezi* hybrid population and any single *X. birchmanni* × *X. malinche* population.
(TIFF)

**S13 Fig. We identified minor parent islands and deserts using genotypes from ancestry
informative sites across the entire genome (referred to as the "unthinned" dataset in the
main text).** To ensure that minor parent islands and deserts were not generated as an artifact
of variation in power to call ancestry along the genome, we re-calculated average ancestry in
these regions using ancestry posterior probabilities generated from an input set of ancestry
informative markers that were thinned to reduce power differences between different regions
of the genome (see Methods; *Local ancestry inference in X. birchmanni* × *X. cortezi hybrids*).
We found few differences in minor parent ancestry in deserts (**A**) based on this analysis. We
identified more variation in ancestry in minor parent islands in the thinned data (**B**), and
excluded a subset of these islands from further analysis (see Methods).
(TIFF)

**S14 Fig.** Shared minor parent ancestry deserts and islands do not have an excess of coding (A) or conserved (B) basepairs compared to other regions of the genome that were not shared ancestry outliers. Gray distributions show number of coding and conserved basepairs in each 0.05 cM window across the genome. Red lines show the median number of coding or conserved basepairs in the 0.05 cM window that is the midpoint of the shared minor parent ancestry deserts. Blue lines show the median number of coding or conserved basepairs in the 0.05 cM window that is the midpoint of the shared minor parent ancestry islands.
(TIFF)

**S15 Fig.** Evaluation of density of repetitive elements (A) and ancestry informative sites (B) in minor parent ancestry deserts and islands relative to the genome-wide background. Distribution in gray shows 0.05 cM windows genome wide, red line shows the median value for the 0.05 cM window that is the midpoint of the shared minor parent ancestry desert, and blue line shows the median value for the 0.05 cM window that is the midpoint of the shared minor parent ancestry islands.
(TIFF)

**S16 Fig. Ancestry in minor parent islands compared to regions of low constraint.** The blue distribution shows minor parent ancestry in the Santa Cruz population in 10 kb windows that are greater than 100 kb from the nearest coding basepair and with an inferred recombination rate in the upper 50% quantile of the genome-wide distribution. The red line shows the average minor parent ancestry in minor parent islands and the gray dashed lines shows the 95% confidence intervals. Thus, in addition to harboring a typical number of coding and conserved basepairs, minor parent islands are still ancestry outliers when compared to regions of the genome expected to have especially low constraint.
(TIFF)

**S17 Fig. Populations in which *X. birchmanni* is the minor parent have a large ~1 Mb desert of *X. birchmanni* ancestry on chromosome 21, the putative sex chromosome.** Plotted here is minor parent ancestry in 10 kb windows relative to average minor parent ancestry genome-wide. Green indicates data from *X. birchmanni* × *X. cortezi* populations and red indicates data from *X. birchmanni* × *X. malinche* populations.
(TIFF)

**S18 Fig. MUMMER alignments indicate that the *X. birchmanni* and *X. cortezi* genomes are co-linear with the exception of several small inversions and the large inversions on chromosome 21 and chromosome 24 shown here.** Alignments shown here also include chromosomes where a shared minor parent desert or island was found to overlap with an inversion.
(TIFF)

**S19 Fig. Each parental species in our analysis differs from other species in a number of structural rearrangements and phylogenetic analysis allowed us to determine whether these rearrangements are likely derived in a particular species (see Methods).** Here we plot minor parent ancestry at inversions that are derived in the major versus minor parent in each independent hybrid population. We find that inversions have unexpectedly high minor parent ancestry regardless of their origin in all hybrid populations, and in several populations inversions derived from the major parent are at unexpectedly low frequencies. Semi-transparent dots show ancestry at individual inversions, solid points and whiskers show the mean ancestry ± 2 standard errors of the mean. Gray line shows average minor parent ancestry in that population genome-wide.
(TIFF)

**S20 Fig. Example of segregation distortion in F2 hybrids between X. birchmanni and X. malinche.** Given the cross design of an $F_1$ intercross we expect 50–50 segregation for parental ancestry types. Indeed, genome-wide average ancestry is 50.3% *X. malinche*. Plotted here is average ancestry by site along chromosome 8. Chromosome 8 has two regions that fall outside of the 99% confidence intervals for ancestry in the cross (shown by the blue shading). (TIFF)

**S21 Fig. X. cortezi ancestry in X. birchmanni × X. cortezi hybrid populations as a function of distance to segregation distorters identified in X. birchmanni × X. malinche early generation hybrids, excluding the segregation distorter on chromosome 6 that is also a shared ancestry desert with low minor parent ancestry across all hybrid populations.** (TIFF)

**S22 Fig. Local ancestry summarized in 10 kb windows for all hybrid populations across the region on chromosome 13 associated with the shared ancestry desert on chromosome 6.** Red–Tlatemaco (*malinche × birchmanni*), Blue–Acuapa and Aguazarca (*birchmanni × malinche*), Green–Santa Cruz (*birchmanni × cortezi*). (TIFF)

**S23 Fig. All hybrid populations included in this study were fixed for the mitochondrial haplotype derived from the majority parental species.** Shown here are the genome-wide ancestry distributions for individuals in *X. birchmanni × X. cortezi* hybrid populations (**A**) and *X. birchmanni × X. malinche* hybrid populations (**B**). Dotted lines show the ancestry of the mitochondrial haplotype for which the population is fixed. (TIFF)

**S24 Fig. Potential interactions between the chromosome 5 shared ancestry desert and two other regions of the genome (chromosome 9 and chromosome 24).** Association between chromosome 5 desert and chromosome 9 and chromosome 24, detected at a FPR of 10% (red line). The FPR 5% threshold is also shown (blue line). Several known gene interactions exist between these three regions: chromosome 24 and chromosome 5—*RBM43* and *trim25*, *rnd3b* and *rasal3*, chromosome 9 and chromosome 24—*prrx1a* and *ccnt2b*. (TIFF)

**S25 Fig. Two possible routes through which Dobzhansky Muller Incompatibilities (DMIs) may arise between diverging lineages. A**. As typically depicted, a derived mutation may arise in each lineage (A and B) which has the potential to negatively interact in hybrids. **B**. DMIs may also arise between the ancestral genotype (denoted as x alleles) and derived alleles that have accumulated on one lineage. This latter scenario may be a possible route through which shared hybrid incompatibilities accumulate between related species, if one lineage has fixed several substitutions and others retain the ancestral genotype. (TIFF)

**S26 Fig. In the main text, we identify segregation distorters based on local ancestry data from F2 hybrids generated between X. birchmanni × X. malinche that deviate from the expected 50–50 ancestry frequency at a given locus.** We performed simulations to ask what selection coefficients are consistent with the deviations from expected admixture proportions that we observe at segregation distortion loci. Shown here is the distribution of accepted selection coefficients from simulations (prior *s* 0–1); see Methods for simulation descriptions. (TIFF)

**S27 Fig. Although we see high X. cortezi ancestry in Huextetitla and Santa Cruz near sites that are strongly selected in X. birchmanni × X. malinche hybrids (Fig 3), we do not see a signature of higher X. cortezi ancestry near putative segregating DMIs identified in natural X. birchmanni × X. malinche hybrid populations.** Shown here are the results for the Santa Cruz population as a function of distance to these sites. Gray lines show results of 500 replicates bootstrap resampling the data, blue shows the average across simulations. (TIFF)

## Acknowledgments

We thank Stepfanie Aguillon, Yaniv Brandvain, Erin Calfee, Yuki Haba, Priya Moorjani, Rachel Moran, Emilie Richards, Ken Thompson, Leslie Turner, and members of the Brandvain, Coop, Martin, Moorjani, Goldberg and Schumer labs for helpful discussion and/or feedback on earlier versions of this work. We also thank Baruc Zago-Mazzocco for field work support. We are grateful to the Mexican federal government for permission to collect samples. We thank Stanford University and the Stanford Research Computing Center for providing computational support for this project.

## Author Contributions

**Conceptualization:** Quinn K. Langdon, Daniel L. Powell, Molly Schumer.

**Data curation:** Quinn K. Langdon, Daniel L. Powell, Shreya M. Banerjee, Paola Fascinetto-Zago, Molly Schumer.

**Formal analysis:** Quinn K. Langdon, Bernard Kim, Cheyenne Payne, Tristram O. Dodge, Ben Moran, Molly Schumer.

**Investigation:** Quinn K. Langdon, Daniel L. Powell, Bernard Kim, Shreya M. Banerjee, Cheyenne Payne, Tristram O. Dodge, Ben Moran, Paola Fascinetto-Zago, Molly Schumer.

**Methodology:** Quinn K. Langdon.

**Project administration:** Molly Schumer.

**Supervision:** Molly Schumer.

**Writing – original draft:** Quinn K. Langdon, Molly Schumer.

**Writing – review & editing:** Quinn K. Langdon, Daniel L. Powell, Bernard Kim, Shreya M. Banerjee, Cheyenne Payne, Tristram O. Dodge, Ben Moran, Molly Schumer.

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
