## [Decision Letter · Decision Letter 0]

16 Aug 2021

Dear Dr Langdon,

Thank you very much for submitting your Research Article entitled 'Predictability and parallelism in the contemporary evolution of hybrid genomes' to PLOS Genetics.

The manuscript was fully evaluated at the editorial level and by independent peer reviewers. The reviewers appreciated the attention to an important topic but identified some concerns that we ask you address in a revised manuscript.

We therefore ask you to modify the manuscript according to the review recommendations. Your revisions should address the specific points made by each reviewer.

[LINK]

Yours sincerely,

Alex Buerkle

Associate Editor

PLOS Genetics

Bret Payseur

Section Editor: Evolution

PLOS Genetics

This manuscript has been reviewed carefully by three referees, each of whom notes the potential contribution of this work to our understanding of repeatability of hybrid genome composition following admixture. The reviews provide a number of thoughtful requests for improvement of the manuscript, including new analyses. Please address each point in your revisions. Concerns about the fit of the demographic model deserve special consideration.

Reviewer's Responses to Questions

**Comments to the Authors:**

Reviewer #1: In this manuscript, Langdon et al. describe and compare patterns of ancestry across multiple admixed populations resulting from hybridization between X. birchmanni and X. cortezi or X. malinche. This results are put in the context of the nature and determinants of genome stabilization following hybridization and what we can learn about the repeatability and predictability of evolution from this. The key results of the paper are (i) that patterns of ancestry are at least somewhat repeatable (more so then expected from neutral evolution), (ii) that repeatability stems in part from selection against minor parent ancestry which interacts with variation in recombination rates across the genome, and (iii) that repeatability also likely reflects the effects of specific incompatibility loci, some of which are identified with some confidence. I found the manuscript clear and interesting overall (i.e., I much enjoyed reading it). I was generally happy with the thoroughness of the analyses, which I think mostly support the authors' claims. With that said, I do have some suggestions for further improving the manuscript.

Main comments:

1. My biggest concern relates to the demographic models, specifically, I am not convinced they are generally a good fit for the data and thus that they justify rejecting a neutral demographic model. I do think there is other good evidence in general for rejecting such a hypothesis, my issue is simply with using the demographic models as justification for this. First, it would be useful to know how well one can estimate the model parameters from the summary statistics used. For this, you can treat simulated data sets (one at a time) as the observed data and estimate the (known) parameters for each. I think it is dangerous to use ABC approaches without doing this (and maybe this was done in a previous paper, but if so say so). Second, some of the priors are pretty constrained, which is especially important to note in cases where the posterior shifts little from the prior. Also, for some, it probably makes more sense to use a log uniform rather than uniform prior so that it is uniform across orders of magnitude (otherwise almost all of the prior probability will be on larger values). Thirds, (which relates back to my first point), is it surprising that the demographic model that was fit without information on the variation in ancestry across the genome being used as a summary statistic fails to capture that aspect of the data? What happens if you include this as a summary statistic.

2. As far as I can tell, the recombination map used was inferred from patterns of LD in nature and thus reflects the composite parameter capturing recombination and Ne. Assuming this is true, the variation in "recombination" across the genome is really variation in recombination and selection across the genome. This makes it a bit trickier to interpret the correlations between recombination rate and ancestry at the local scales (i.e., across windows). I think this is mostly fine, but is worth acknowledging.

Other minor comments:

L170-180, I didn't see where a bottleneck model was considered? From what I can tell the ABC model only specified a single hybrid population size not an initial bottleneck population size followed by a rebound.

L242, This is subjective, but I don't know that I would call rho values of 0.25 and 0.17 "surprisingly high". Rank correlations are perhaps a bit tricky to think about, but it we treat these like Pearson correlations and then calculate r2 from that, this would suggest that patterns of minor parent ancestry in one species explain ~3% (0.17^2 = 0.0289) to ~6% (0.25^2 = 0.0625) of the variation in patterns of ancestry in the other. I think saying something like that would better convey the results.

L298, Probably should change "indicate" to "consistent with".

L350, Typo, "cross" to "across".

Table 1, Is there a mistake here with the dN/dS ratios or do they really span ~6 orders of magnitude (i.e., 0.001 to 102). IN particular, maybe 102 and 98 should be 1.02 and 0.98 or something like that.

L421, Others have considered predictability and repeatability in the context of hybridization and even specifically genome stabilization. See in particular Rieseberg et al. 2003 (10.1126/science.1086949) (which is cited but should be mentioned here and in this specific context) and related work on sunflowers and Chaturvedi et al. 2020 (https://doi.org/10.1038/s41467-020-15641-x).

Fig 2 B, This should be Spearman rho not Pearson R I think, and what is the Pearson correlation (or why use rank correlations for everything here, consider reporting both).

Reviewer #2: This is a very nice paper on an amazing study system. The authors examine whether the patterns of genome ancestry in hybrids are repeatable among replicated populations of hybrids from the same parental species and among different sets of parental species. This seems to be high quality and rigorous work. I have only a few comments:

The paper is densely written and refers to a lot of supplementary material, which makes it hard to read and follow at some places and a difficult paper to review. Also, it would be useful in the results to detail the method since the method sections comes after the results. Including more details on the assumptions behind the analyses and simulations would also help. For instance, on line 251, the authors say ‘after accounting for shared genomic architecture, we still’ . The accounting for the shared architecture seems a very critical analysis here since the signal left is attributed to selection. It would be useful to know how this accounting was done and how robust it is to all the assumptions that the authors have to put into it.

Figure 2A: I imagine that the windows are non-overlapping windows?

Figure 3A: y-axis could be better labeled. A color legend for the grey/colored points would be useful.

In the analyses where there is a relationship between conserved base pairs and ancestry of particular windows. I wonder if the accuracy of the ancestry calling also depends on the level of nucleotide divergence in these windows. Since priors are used to assign the ancestry, a weak signal due to few SNPs for calling the ancestry may be more affected by the priors and favor one parent or the other?

Line 242: Re the correlation of ancestry across the genome. I was wondering if windows are non-overlapping to make sure the correlation tests are valid.

I am not sure I understand the data presented in Table 1. Is the Dn/Ds ratio shown 102? From the data described it seems that there is no synonymous change so a ratio should not be calculated because it would be infinite? It would be useful to discuss more the observations and also the quality of the mapping data in these regions to make sure they are clearly a single-copy gene etc.

Line 685: it is not clear what the simulations to examine the correlations of ancestry take into account, for instance the presence of inversions in the parental genomes.

Reviewer #3: This paper presents a very thorough exploration of how admixture landscapes are correlated with various genomic features and between different hybrid swordtail fish populations. The authors examine several hybrid populations with different “major” and “minor” parent species (i.e. the parent that contributes the majority of the gene pool versus the minority of the gene pool) and broadly explore the properties of the regions of the genome that contain less minor parent ancestry than expected by chance. Most of the correlations of minor parent ancestry depletion presented in the paper (e.g. with recombination rate, linkage to conserved base pairs) have been hypothesized in the past and demonstrated in other hybrid systems, but it is valuable to see these patterns corroborated in a set of independent hybrid zones being analyzed in a rigorous and consistent way.

One of the main findings of the paper are shared deserts of minor parent ancestry among parallel swordtail fish hybrid zones, a result that is reminiscent of the overlap in deserts previously identified between Neanderthal and Denisovan introgression landscapes in humans. I think this result would be strengthened if the authors presented a clearer argument for the claim in line 280 that “sharing of both low and high minor parent ancestry regions exceeded sharing expected by chance (Fig 3A; Methods, Supporting Information 7; p<0.001 by simulation).” I couldn’t figure out how this p value was calculated exactly, or whether it suffers from an error commonly made when computing correlations between time series data. Like a traditional time series, the proportion of minor parent ancestry in each population varies continuously along the genome, which implies that if it happens to be low in the same region of two landscapes at once, it will also be low in some surrounding window, and these datapoints cannot be regarded as independent samples due to the continuity of the data. When quantifying the correlation between minor parent ancestry deserts, and especially when assigning it a p value, the authors should take care to avoid this issue (described in more detail here https://academic.oup.com/mbe/article/23/5/911/1058377).

From reading supporting information 7, I think this claim of minor parent desert overlap is based on a comparison to a null consisting of simulated populations that were each picked to have negative selection against minor parent ancestry at 1 random site. This seems much too simple to reproduce features like the correlation of recombination rate with minor parent ancestry, since this would require selection against minor parent ancestry at much more than 1 site per genome. Later on in the manuscript the authors clarify that the minor parent ancestry deserts are identified in a way that corrects for recombination rate, and they also perform a post-hoc test showing that exon density can’t account for the results, but I think the simulation they initially compare to is enough of a straw man that it would be more useful to just cite these latter results rather that cite a p-value based on a comparison to genomes where selection acts on just 1 site each.

Although this is not essential, I also think the manuscript could be improved by adding more analyses to the Tlatemaco population, which is noted in a few cases to be an outlier. The authors hypothesize that Tlatemaco behaves differently than the others because the minor parent has higher effective population size than the major parent, which could mean that selection for lower genetic load works in opposition to selection to minimize hybrid incompatibilities. Is this hypothesis testable by digging more into where the Tlatemaco landscape exhibits similarities and differences to the other birchmanni x malinche populations? For example, does Tlatemaco display the expected behavior near segregation distorters, which probably aren’t too sensitive to effective population size? What about the other candidate incompatibility loci? Are genes under the most purifying selection like housekeeping genes the ones where this population might display evidence for selection against the major parent ancestry due to genetic load?

Minor comment:

“principal components” is misspelled throughout supporting information section 2

**Have all data underlying the figures and results presented in the manuscript been provided?**

Reviewer #1: Yes

Reviewer #2: Yes

Reviewer #3: Yes

PLOS authors have the option to publish the peer review history of their article (what does this mean?). If published, this will include your full peer review and any attached files.

Reviewer #1: No

Reviewer #2: No

Reviewer #3: No

---

## [Editor Report · Decision Letter 1]

28 Oct 2021

Dear Dr Langdon,

We are pleased to inform you that your manuscript entitled "Predictability and parallelism in the contemporary evolution of hybrid genomes" has been editorially accepted for publication in PLOS Genetics. Congratulations!

Yours sincerely,

Alex Buerkle

Associate Editor

PLOS Genetics

Bret Payseur

Section Editor: Evolution

PLOS Genetics

Comments from the reviewers (if applicable):

The authors have added substantially to the analyses and presentation in the manuscript and have thoroughly responded to the suggestions and questions from the previous round of review. I appreciate the care and clarity of the responses.

One small point arose in my reading of the manuscript: on line 412 the manuscripts indicates that incompatibilities by definition involve more than one locus. This is not quite correct. Epistatic incompatibilities require two or more loci. But underdominance and heterozygote inferiority exist and are surely important and highly effective components of reproductive isolation. I do not think the references to incompatibilities need to be changed throughout, but on line 412 and perhaps elsewhere it would be helpful to clarify that the manuscript is referring to incompatibilities involving epistasis.

Overall, in addition to the evidence it provides for some predictability and determinism in outcomes of hybridization, I think this manuscript will be a great point of reference for care in modeling and methods to account for possible causes of observed patterns.

**Data Deposition**

http://datadryad.org/submit?journalID=pgenetics&manu=PGENETICS-D-21-01004R1

**Press Queries**

---

## [Editor Report · Acceptance letter]

30 Nov 2021

PGENETICS-D-21-01004R1 

Predictability and parallelism in the contemporary evolution of hybrid genomes 

Dear Dr Langdon, 

We are pleased to inform you that your manuscript entitled "Predictability and parallelism in the contemporary evolution of hybrid genomes" has been formally accepted for publication in PLOS Genetics! Your manuscript is now with our production department and you will be notified of the publication date in due course.

With kind regards,

Andrea Szabo

PLOS Genetics

On behalf of:
